# Sources and nature of ice-nucleating particles in the free troposphere at Jungfraujoch in winter 2017

Larissa Lacher[1,3], Hans-Christian Clemen[2], Xiaoli Shen[3,a], Stephan Mertes[4], Martin Gysel-Beer[5], Alireza Moallemi[5], Martin Steinbacher[6], Stephan Henne[6], Harald Saathoff[3], Ottmar Möhler[3], Kristina Höhler[3], Thea Schiebel[3], Daniel Weber[7,b], Jann Schrod[7], Johannes Schneider[2], and Zamin A. Kanji[1]

[1]Department of Environmental System Sciences, Institute for Atmospheric and Climate Science, ETH Zurich, 8092, Zurich, Switzerland

[2]Particle Chemistry Department, Max Planck Institute for Chemistry, 55128 Mainz, Germany

[3]Institute of Meteorology and Climate Research, Karlsruhe Institute of Technology, 76344, Eggenstein-Leopoldshafen, Germany

[4]Leibniz Institute for Tropospheric Research, 04318, Leipzig, Germany

[5]Laboratory of Atmospheric Chemistry, Paul Scherrer Institute, 5232, Villigen, Switzerland

[6]Empa, Swiss Federal Laboratories for Materials Science and Technology, 8600, Duebendorf, Switzerland

[7]Institute for Atmospheric and Environmental Sciences, Goethe University Frankfurt, 60438 Frankfurt, Germany

[a] now at: Department of Earth, Atmospheric, and Planetary Sciences, Purdue University, 47907, West Lafayette, Indiana, United States

[b] now at: Federal Waterways Engineering and Research Institute, 76187, Karlsruhe, Germany

*Correspondence to*: Larissa Lacher (larissa.lacher@kit.edu), Zamin A. Kanji (zamin.kanji@env.ethz.ch)

**Abstract**. Primary ice formation in mixed-phase clouds is initiated by a minute subset of the ambient aerosol population, called ice-nucleating particles (INPs). The knowledge about their atmospheric concentration, their composition, and source in cloud-relevant environments is still limited. During the joint INUIT/CLACE (Ice Nuclei research UnIT/ CLoud–Aerosol Characterization Experiment) 2017 field campaign, observations of INPs, as well as of aerosol physical and chemical properties were performed, complemented by source region modelling. This aimed at investigating the nature and sources of INPs. The campaign took place at the High-Altitude Research Station Jungfraujoch (JFJ), a location where mixed-phase clouds frequently occur. Due to its altitude of 3580 m a.s.l., the station is usually located in the lower free troposphere, but can also receive air masses from terrestrial and marine sources via long-range transport. INP concentrations were quasi-continuously detected with the Horizontal Ice Nucleation Chamber (HINC) under conditions representing the formation of mixed-phase clouds at -31 °C. The INP measurements were performed in parallel to aerosol measurements from two single particle mass spectrometers, the Aircraft-based Laser ABlation Aerosol MAss Spectrometer (ALABAMA) and the Laser Ablation Aerosol Particle Time-Of-Flight mass spectrometer (LAAPTOF). The chemical identity of INPs is inferred by correlating the time series of ion signals measured by the mass spectrometers with the time series of INP measurements. Moreover, our results are complemented by the direct analysis of ice particle residuals (IPRs) by using an ice-selective inlet (Ice-CVI) coupled with the ALABAMA. Mineral dust particles and aged sea spray particles showed the highest correlations with the INP time series. Their role as INP is further supported by source emission sensitivity analysis using atmospheric transport modelling, which confirmed that air masses were advected from the Saharan desert and marine environments during times of elevated INP concentrations and ice-active surface site densities. Indeed, the IPR analysis showed that by number, mineral dust particles dominated the IPR composition (~58%), and also biological and metallic particles are found to a smaller extent (~10% each). Sea spray particles are also found as IPRs (17%), and their fraction in the IPRs strongly varied according to the increased presence of small IPRs, which is likely due to an impact from secondary ice crystal formation. This study shows the capability of combining INP concentration measurements with chemical characterization of aerosol particles using single particle mass

spectrometry, source region modelling, and analysis of ice-residuals in an environment directly relevant for mixed-phase cloud formation.

## 1 Introduction

Ice-nucleating particles (INPs) are a rare subset of the ambient aerosol particle population (e.g. Rogers et al., 1998; DeMott et al., 2010) and are an important atmospheric constituent since they can modulate the microphysical properties of cirrus and mixed-phase clouds (MPCs) by initiating ice crystal formation. INPs have the ability to change the sensitive balance between the liquid and ice water content of MPCs, which can lead to rapid cloud glaciation and associated dissipation (e.g. Lohmann et al., 2002; Sassen et al., 2003; Fu et al., 2017; Desai et al., 2019), or to cloud brightening and related changes in the radiative properties (Solomon et al., 2018). Moreover, precipitation is formed efficiently via the ice phase (Pruppacher and Klett, 1997; Mülmenstädt et al., 2015; Field and Heymsfield, 2015; Heymsfield et al., 2020). In the absence of efficient secondary ice formation processes (Korolev and Leisner, 2020), ice formation induced by INPs is directly relevant for the Earth's radiation and water budget, and even relevant for the initiation of secondary ice processes. Despite their importance, the knowledge about the abundance and nature of INPs in the atmosphere still needs to be improved, in part because of their low ambient concentration, and high spatio-temporal variability. Ambient INP concentrations were found to range between less than $10^{-5}$ stdL$^{-1}$ to greater than $10^3$ stdL$^{-1}$ between -10 °C and -35 °C, respectively (Kanji et al., 2017). While their ambient concentration is a strong function of ice nucleation temperature, variations over several orders of magnitude are reported at any one temperature, implying the importance of the ice nucleation ability of individual aerosol particle types. Depending on temperature range as well as atmospheric abundance, several aerosol particle types might dominate the INP population. Mineral dust is recognised as a key ice nucleator in the troposphere at temperatures below -15 °C (DeMott et al., 2003; Cziczo et al., 2003; Richardson et al., 2007; Chou et al., 2011; Atkinson et al., 2013; Tang et al., 2016; Boose et al., 2016a, b; Jiang et al., 2016; Kanji et al., 2017; Price et al., 2018; Welti et al., 2018). However, in the absence of highly ice active material or at warmer temperatures, other aerosol particle types are important. Aerosol particles emitted from the ocean were identified to be ice active at temperatures below -5 °C (e.g. Brier and Kline, 1959; Bigg, 1973; Schnell 1977; Wilson et al., 2015; DeMott et al., 2016; Veragra-Temprado et al., 2017; McCluskey et al., 2018) and showed their potential to impact cloud properties in remote marine environments (Huang et al., 2018; Vergara-Temprado et al., 2018; McCluskey et al., 2019). Also, primary biological particles might contribute significantly to the INP population, especially at temperatures warmer than -15 °C (Pratt et al., 2009; Schmidt et al., 2017; O'Sullivan et al., 2018), but their atmospheric contribution on a global scale is still unclear (Hoose et al., 2010; Burrows et al., 2013; Sesartic et al., 2013). Moreover, metallic particles might act as ambient INPs at MPC conditions (Cziczo et al., 2009; Kamphus et al., 2010; Ebert et al., 2011; Worringen et al., 2015) as well as aerosol particles emitted by anthropogenic sources, such as from combustion processes, however, contradictory results exist (Cozic et al., 2008; Kupiszeswski et al., 2016; Mahrt et al., 2018; Kanji et al., 2020).

Next to laboratory experiments using a specific aerosol particle type representative for ambient conditions, attempts to directly identify INPs in ambient air are made by sampling in air masses which are dominated by one aerosol particle type, such as. Saharan dust particles (DeMott et al., 2015; Boose et al., 2016a), marine aerosols (e.g. Bigg, 1973; McCluskey et al., 2018; Irish et al., 2019; Ladino et al., 2019), biological aerosols (e.g. Huffman et al., 2013; Mason et al., 2015), or aerosols from anthropogenic sources in an urban environment (e.g. Chen et al., 2018). Nevertheless, this approach is limited since it is not guaranteed that just a single aerosol particle type is present in ambient air and at relevant concentrations, which is crucial in the light of the very low fraction of ice active particles. An indirect approach is the investigation of relationships between INP concentrations and parallel-measured parameters of aerosol particle and air mass properties (e.g. Mason et al., 2015; Boose et al., 2016; Lacher et al., 2018b), which can result in useful parameterizations of INPs (e.g. DeMott et al., 2010). For example, the chemical composition of aerosol particles can be investigated in parallel to INP measurements with a single particle mass spectrometer (SPMS; Murphy et al., 2007; Chen et al., 2021). In addition, an SPMS together with a pumped counterflow

virtual impactor (PCVI; e.g. DeMott et al., 2003; Hiranuma et al., 2016) can also be connected to a continuous-flow-diffusion chamber (CFDC) and thus directly analyze the activated INPs chemically (e.g. Cziczo et al., 2003; Richardson et al., 2007; Cornwell et al., 2019). The SPMS is a common tool in the field of ice nucleation research (Cziczo et al., 2017) and its use led to an improved understanding of ice formation in clouds (Cziczo et al., 2004; Pratt et al., 2009; Cziczo et al., 2013; Cziczo and Froyd, 2014; Lin et al., 2017; Cornwell et al., 2019). Another direct method to identify INPs is the application of ice-selective inlets (e.g. Mertes et al., 2007), which sample freshly nucleated ice crystals in clouds, sublimate them and the resulting initial INP are analyzed with different methods as e.g. single-particle mass spectrometry and scanning electron microscopy (Cziczo et al., 2009; Kamphus et al., 2010; Schmidt et al. 2017, Eriksen Hammer et al., 2018).

Here, we present results from an intensive field campaign conducted at a high-altitude site during January and February 2017. A suite of instruments to detect INPs, as well as two SPMSs to characterize aerosol particle composition were applied in combination with measurements with an ice-selective inlet, aiming at getting better insights into INP characteristics and sources, and aerosol particle properties relevant for MPC formation.

## 2 Methods

### 2.1 Location and overview of the field campaign

The INUIT/CLACE 2017 (Ice Nuclei research UnIT/ CLoud–Aerosol Characterization Experiment) field campaign took place from January 22 - February 22, 2017 at the High Altitude Research Station Jungfraujoch (JFJ), as a joint activity of international research groups to study aerosol-cloud interactions (e.g. Eriksen Hammer et al., 2018). Being located in the Swiss Alps at an altitude of 3580 m a.s.l. (46°33'N, 7°59'E), JFJ is an outstanding location for such a purpose, since it is not only a natural environment for cloud formation and occurrence (e.g. Herrmann et al., 2015), but also because it receives air masses of different origins (Cui et al., 2011). Due to its elevation, the site is usually exposed to the lower free troposphere (> 60% of the time in winter; Herrmann et al., 2015), where the larger fraction of the aerosol particle population (accumulation and coarse mode particles > 90 nm; Herrmann et al., 2015) are not formed in-situ or from local point sources, but are only present at the site due to long-range transport. This is confirmed by analysis of trace gases such as the ratio of reactive nitrogen species ($NO_y$) to carbon monoxide (CO), which can be used as an indicator for air mass age (Zellweger et al., 2003). Therefore, JFJ is a site representative for detecting particles from various source regions being transported within the free troposphere. However, the aerosol particle measurements can be impacted in the short-term by daytime tourist activities. Air masses that originate from the Saharan desert frequently reach the site which results in strongly elevated mineral dust loads (Collaud Coen et al., 2004; Chou et al., 2011; Boose et al., 2016; Lacher et al., 2018). Also, air masses originating in the marine boundary layer reach JFJ (Cui et al., 2011; Lacher et al., 2018), particularly in the warm season, convectively lifted boundary layer air can reach the site (e.g. Collaud Coen et al., 2011), injecting air masses from local and regional sources such as e.g. forests or anthropogenic polluted environments. Hence, at the JFJ a large variety of aerosol particle types can be found ranging from relatively fresh emissions from the polluted boundary layer (typically in the summer) to aged aerosol particles from different regions as transported in the free troposphere over large distances. This site is therefore ideal to study the impact of different aerosol populations on cloud properties.

JFJ is part of the Global Atmospheric Watch (GAW) monitoring programme as well as of the Aerosol, Clouds and Trace Gases Research Infrastructure (ACTRIS), the Swiss National Air Pollution Monitoring Network (NABEL) and the SwissMetNet meteorological network. As such, important parameters of aerosol particle physical properties (Baltensperger et al, 1997; Herrmann et al., 2015; Bukowiecki et al., 2016), as well as trace gases are routinely monitored (Steinbacher et al., 2020) and accompanied by measurements of meteorological parameters (Appenzeller et al., 2008). Some of the routine observations like the reactive nitrogen to carbon monoxide ratio and the wind direction are used for the interpretation of the results below. One important parameter is the aerosol particle size distribution, which is measured using a scanning mobility particle sizer (SMPS;

0.02 – 0.6 µm), consisting of a differential mobility analyzer (DMA; TSI 3071, TSI Inc. Shoreview, USA) and a condensation particle counter (CPC; TSI 3775, TSI Inc. Shoreview, USA), and an optical particle sizer (OPS; 0.3 – 10 µm; TSI 3300, TSI Inc. Shoreview, USA). Aerosol particle measurements are conducted at the GAW total aerosol inlet (Weingartner et al., 1999), as well as at an ice selective inlet (Ice-CVI; Mertes et al., 2007) as depicted in Fig. 1 and described in more detail in the following sections.

## 2. 2 INP measurements

### 2.2.1 Online INP measurements

The number concentration of INPs ([INP]) was determined using the Horizontal Ice Nucleation Chamber (HINC; Lacher et al., 2017) at -31 °C (±0.4 °C; [INP]$_{-31}$) and at a relative humidity with respect to water (RH$_w$) of 103% (±2%), representing condensation/immersion freezing, relevant for the formation of MPCs (e.g. DeBoer et al., 2011; Murray et al., 2012; DeMott et al., 2015; DeMott et al., 2018). A nucleation temperature of -31 °C was chosen in order to avoid measurements below the detection limit of the instrument. This is crucial at a remote location as JFJ where the [INP] is naturally low, but is generally higher at colder nucleation temperatures. Moreover, a nucleation temperature of -31 °C ensures that a large spectrum of INP activation at MPCs is covered. The RH$_w$ of 103% ensures that the entire aerosol layer which experiences a varying RH between 101 – 103% is above water saturation such that the particles can activate into droplets in the given residence time of HINC. Cloud-like temperature and saturation conditions in HINC are established by applying a temperature gradient along ice-coated parallel plates, held at sub-zero temperatures. INPs are sampled into the chamber guided within a sheath flow and exposed to the temperature and supersaturation in the center of the chamber and can activate into water droplets and/or ice crystals. Due to particle losses in the system, mainly due to the tubing upstream of the chamber and its horizontal orientation, the measurements are representative for aerosol particles < 2 µm (56% transmission efficiency at this size), which is the dominant size range for the aerosols present at the site (Baltensperger et al., 1997; Nyeki et al., 1998). We note that in this way some of the larger (> 2 µm) particles may not be detected. For more details on the transmission efficiency see Lacher et al. (2017). [INP]$_{-31}$ was measured with a time resolution of 20 minutes, which are alternated by 10 minutes of background measurements via a filter to exclude impact of counting undesired frost particles generated within HINC. The instrument's limit of detection is thereby calculated based on these background measurements. To achieve better measurement statistics due to the naturally low [INP] in the free troposphere, an aerosol particle concentrator (the Portable Fine Particle Concentrator (PFPC; Gute et al., 2019) was deployed upstream of HINC during the field campaign, allowing an enrichment in aerosol particles > 0.1 µm. The enrichment is thereby size-dependent due to the working principle of the PFPC, with an enrichment factor of ~ 10 at particle sizes of 0.3 µm, a maximum enrichment of ~20 for particles > 0.75 µm (Gute et al., 2019). The INP enrichment factors were determined by consecutive measurements on and off the concentrator, and showed a large variability between values of 1 and 23, reflecting the variability in the size of the present INP population (see Lacher et al., 2018a for a more detailed description of this setup). For comparison and correlation with other measurements, only quantifiable [INP]$_{-31}$ > limit of detection (~ 0.2 stdL$^{-1}$) were considered for this study. Here we present results from [INP]$_{-31}$ and its related ice-active surface site density ($n_s$), which normalizes [INP] to the available surface area per volume of air of the ambient aerosol particles (Connolly et al., 2009; Hoose and Möhler, 2012):

$$n_s = \frac{[INP]\,(\# \, m^{-3})}{total\ particle\ surface\ area\ (m^2 m^{-3})} \tag{1}$$

given in m$^{-2}$, and which requires the surface area distribution concentration to be calculated from the number size distribution. Measurements from the GAW SMPS and OPS are used to calculate the available surface area of the aerosol population by assuming a uniform particle shape and that the refractive index of the ambient aerosol population is represented by the calibrated value of the OPS. We acknowledge that this can lead to higher uncertainties, which are not quantified here. The concept of $n_s$ is based on the assumption of a uniform composition of the investigated aerosol sample, and assumes that the

temperature dependence of ice nucleation is more important than the time dependence, which therefore can be neglected (e.g., Welti et al., 2012).

In order to investigate the relationship between [INP]$_{-31}$ and $n_s$ with meteorological and aerosol parameters, we use Spearman's rank correlation coefficient (Spearman, 1904). The Spearman's rank correlation determines to what extent two variables are monotonically related by applying a linear correlation analysis on the rank-ordered values of the parameters. As we would not necessarily await a linear relationship amongst parameters in atmospheric science, this test is well suited for our purposes. Examples for the correlation analysis are presented in Fig. S6.

### 2.2.2 Offline INP measurements

Filter-based aerosol collection followed by offline quantification of [INP] was performed with the FRankfurt Ice nucleation Deposition freezinG Experiment (FRIDGE; Bundke et al., 2008; Klein et al., 2010; Schrod et al., 2016) and the Ice Nucleation Spectrometer of the Karlsruhe Institute of Technology (INSEKT) which is a re-built version of the Ice Spectrometer freezing method (Garcia et al., 2012; Hill et al., 2016).

For FRIDGE, the aerosol particles are sampled on silicon wafers using electrostatic precipitation with a time resolution of 1 – 8 hours. After collection, the wafers were analyzed on site in an isostatic diffusion chamber. The sample is first cooled to the desired sub-zero temperature which is measured with an accuracy of ±0.2 °C. Then, the sample is exposed to a defined amount of water vapor (i.e., here corresponding to 101% RH$_w$), relevant for the formation of MPCs. Subsequently, ice crystals start to grow on INPs, which causes a change in brightness that is detected and recorded by a camera.

The INSEKT samples were collected on precleaned (10% H$_2$O$_2$ solution followed by ultrapure water) polycarbonate filters with pore sizes of 0.2 µm (Whatman plc, Kent, United Kingdom) with a flow of 9.6 liter per minute for 8 hours (daytime) or 12 hours (nighttime). The samples were shipped to the laboratory at the Karlsruhe Institute of Technology for analysis. In brief, the collected aerosol particles are washed off the filter with 8 mL of nanopure water and the resulting suspension is then pipetted into 80 wells of PCR plates, with each well having a volume of 50 µL, together with aliquots from the nanopure water used to create the suspension. The samples are then placed into aluminum holders which are cooled using a cooling bath, measuring down to temperatures of ~ -25 °C at a 0.25 °C min$^{-1}$ cooling rate. The frozen wells are detected optically by a camera due to a brightness change in the frozen aliquots. For each experiment, a background correction is applied, taking into account the freezing of the pure water used to create the suspension.

The calculation for the cumulative [INP] is based on the sampled volume during filter collection, and follows the equation from Vali (1971). Here, we present [INP] at -10 °C, -15 °C and -20 °C for results from INSEKT, and -20 °C, -25 °C and -30 °C for FRIDGE ([INP]$_{-10}$ to [INP]$_{-30}$).

### 2.3 Aerosol composition measurements by mass spectrometry

### 2.3.1 Instrument description

Aerosol particle composition was measured with three different aerosol mass spectrometers: The Laser Ablation Aerosol Particle Time-Of-Flight mass spectrometer (LAAPTOF, AeroMegt GmbH), the Aircraft-based Laser ABlation Aerosol MAss spectrometer (ALABAMA), and a Compact Time-of-Flight Aerosol Mass Spectrometer (C-ToF-AMS).

The LAAPTOF and the ALABAMA are based on similar measurement concepts: Aerosol particles are transmitted from ambient air into a vacuum chamber through an aerodynamic lens which focuses the particles between 0.07– 2.5 µm onto a narrow beam. The particles are then detected by two continuous wave lasers, which allows for measuring the velocity and thereby their vacuum aerodynamic diameter. Particle detection at both detection stages triggers an ablation laser that emits a laser pulse onto the particle, thereby ablating the particle and ionizing its components. The ions are detected in bipolar time-of-flight mass spectrometers. The overall detection efficiency (combining the detection efficiency and the hit rate) of the

LAAPTOF is between 0.01 (±0.01) % to 4.2 (±2.4) %, in the size range of 0.2 to 2 µm based on polystyrene latex particles (PSL). The highest overall detection efficiency is for 1µm and lowest for 2 µm (Shen et al., 2018). Note that such efficiency is also particle type dependent (Shen et al., 2018, 2019). More details on the LAAPTOF can be found in Gemayel et al. (2016), Reitz et al. (2016), and Shen et al. (2018, 2019). Details on the ALABAMA have been presented in Brands et al. (2011), Roth et al. (2016), Schmidt et al. (2017), and Clemen et al. (2020). The detection efficiency of the ALABAMA during this campaign was between 40% and 60% in the size range of 0.3 to 1.0 µm based on PSL particles. Up to a particle size of about 1.3 µm, the detection efficiency of the ALABAMA decreased to less than 30% and is estimated to be about 5 (±5) % for 2 µm. At the same time, the hit rate during those tests using PSL particles was lower, such that the overall detection efficiency for the ALABAMA was only between 1% and 16% in the size range from 0.3 to 1 µm. As those values are based on measurements using PSL particles, they can vary considerably during field applications; e.g., the ALABAMA hit rates were significantly higher than those of the PSL test measurements (which is attributed to particle charge effects during the nebulization of the PSL particles). In the light of our research objectives, focusing on the general trend of the aerosol particle composition, we therefore provide an overview of the size dependent overall detection efficiency from the LAAPTOF and the ALABAMA normalized to the maximum value measured, together with the normalized transmission efficiency from HINC (Fig. S1). From those normalized values it is visible that both SPMSs detect aerosol particles in the same size range, with a maximum between 0.5 and 1 µm, and therefore yield comparable information on the particle composition in this size range. HINC measures particles below 2 µm with a high efficiency which can have an impact on the comparison between the INP measurements from HINC and the results obtained from the ALABAMA and the LAAPTOF.

The data product of both instruments is a bipolar ion signature on a single particle basis. The main difference between LAAPTOF and ALABAMA with respect to the chemical analysis is the laser wavelength that is used for ablation. The LAAPTOF uses an ArF excimer laser with a wavelength of 193 nm (pulse duration: 4 to 8 ns), whereas the ALABAMA uses a quadrupled Nd:YAG with a wavelength of 266 nm (pulse duration: 6 ns). The laser energy per pulse used during this campaign ranged between 3 mJ and 4 mJ for the LAAPTOF and between 7.2 mJ and 9 mJ for the ALABAMA. These differences in laser wavelength and energy result in different ablation and ionization efficiencis that affect the relative ion signals in the particle mass spectra. More specifically, the different lasers lead to different power densities. ALABAMA generates a power density of $1x10^9$ W cm$^{-2}$, with an assumed effective diameter of the laser at the ablation spot of the particles of about 400 µm (Clemen et al., 2020), whereas the LAAPTOF uses a power density of $\sim 1x10^{10}$ W cm$^{-2}$, with a laser beam diameter of 99 (±31) µm at the ablation spot (Ramisetty et al., 2018). The C-ToF-AMS has been described in detail in the literature (e.g., Drewnick et al. 2005, Canagaratna et al., 2007). Its data product is a quantitative mass concentration of non-refractory aerosol compounds, e.g. ammonium, nitrate, sulfate, and organics, in particles with an aerodynamic diameter below 1 µm (PM1), based on averaging over an ensemble of particles. The focus of the analysis presented here lies on the single particle instruments, because they are able to provide information on both non-refractory and refractory particles indicating the presence of mineral dust, elemental carbon, sea spray, metals, or primary biological particles.

**2.3.2 Data analysis**

For an overview of the ambient aerosol properties of the whole measurement period, we used a clustering algorithm to infer the dominating aerosol particle type. This was done using the LAAPTOF data, which covered a longer time period. The ALABAMA was not operated in optimal configuration in the beginning of the campaign because the extraction voltages for the mass spectrometer had to be optimized after the installation of a so-called delayed ion extractor (see Clemen et al. (2020) for more details). Furthermore, during most of the time when the station was in clouds, the ALABAMA was connected to an ice-selective inlet (the Ice-CVI; see section 2.6) to sample freshly formed ice particle residuals. The mass spectra recorded by the LAAPTOF were classified by fuzzy c-means clustering method as described in Reitz et al. (2016) and Shen et al. (2018). Briefly, this classification method attributes individual particles to multiple classes or clusters according to spectral similarities,

typically calculated by linear correlation or Eulerian distance. The particle classes are attributed to particle types based on marker ions that have been identified in reference spectra of particle types.

To show that both the LAAPTOF and the ALABAMA yield comparable results in spite of the different laser wavelengths and energies for ablation and ionization, cation and anion mass spectra from both instruments averaged over 24 hours of parallel sampling are compared (see Supplement Fig. S2). A Spearman's rank correlation between both mass spectra yields a correlation coefficient of 0.81 for cations, and 0.59 for anions. Although many peaks can be identified in both mass spectra, the relative height of many ion signals is different, due to the different ionization laser energy. In general, this suggests a reasonable agreement in the overall detection of aerosol particles between the two instruments.

To evaluate the chemical information from the single particle mass spectrometry against the $[INP]_{-31}$ and $n_s$, we did not use the cluster analysis, because, as explained above, cluster algorithms group mass spectra by similarity. This similarity is calculated by linear correlation or Eulerian distance and is therefore mainly influenced by large ion signals. Thus, small ion signals that might still represent a chemical component that is important for ice nucleation may be overlooked by this method. Instead, we chose to correlate the time series of individual ions with the time series of the $[INP]_{-31}$ and $n_s$ as follows: The HINC INP data represented average values over 20 minutes. To transpose the mass spectrometer data onto these 20-minute intervals, we used two methods. Method 1 counts whether a certain ion is present in a mass spectrum during the 20-minute interval (i.e. above the noise threshold) and reports the fraction of mass spectra that contain this ion. For example, a fraction of 0.1 means that 10% of all mass spectra in this 20 min interval contain this ion. Method 2 averages all mass spectra in this 20 min interval after normalizing the spectra to their total ion signal and reports the relative height of a certain ion in the average mass spectrum. Thus, a fraction of 0.1 means that this ion has a relative signal of 10% with respect to the total ion signal of the 20-min-averaged mass spectrum. Method 1 therefore still contains the single particle information, reflecting the relative abundance of ions on a particle-by-particle basis, whereas method 2 is more representative of the average composition of all particles during this time interval. We note that the sensitivity of the instruments can vary substantially for different ions which can have a significant impact on the apparent average composition.

The resulting time series of all ions up to $m/z$ 250 were then correlated to the time series of the $[INP]_{-31}$ and $n_s$ using Spearman's rank correlation. This was done for the positive ions (cations) and for the negative ions (anions) separately. Non-significant rank correlations (p > 0.05) were excluded. The remaining rank correlation coefficients ($r$) for all ions up to $m/z$ 250 were squared ($r^2$), averaged, and their standard deviation ($\sigma$) was calculated. Only ions whose $r^2$ values were greater than $1\sigma$ above the mean value were selected as meaningful correlators (see Supplement Fig. S3) to $[INP]_{-31}$ or $n_s$. To correlate the peaks from the particle spectra with $[INP]_{-31}$ and $n_s$, we used the non-squared coefficients ($r$) of the selected ions to allow investigation for negative correlations. We note that the instruments used here for the correlation coefficient analysis have different size-dependent particle transmission efficiencies but overlap in the region < 2 µm. While HINC, LAAPTOF and ALABAMA have their maximum transmission/detection efficiency for particles < 1 µm, HINC has a higher transmission efficiency for larger particles (56% for 2 µm particles), as compared to the LAAPTOF and ALABAMA (~0.01% and ~5%, respectively). The advantage of the ion correlation method is that it looks at the correlation of chemical substances rather than whole particle types, which means that fewer initial assumptions have to be made and a cross-particle type approach can be taken.

## 2.4 Fluorescent particles and black carbon

Fluorescent aerosol particles were monitored during the INUIT/CLACE 2017 field campaign using the Wideband Integrated Bioaerosol Sensor (WIBS; model 5/NEO, DMT, Longmont, Colorado, USA) to investigate the potential impact from biological particles. The WIBS detects single particles in the size range of 0.5 - 20 µm and categorises them based on their fluorescence and light scattering properties (e.g. Kaye et al. 2005). Specifically, aerosol size and shape are detected by forward-scattered light from a 635 nm diode laser, and detection of fluorescent particles is achieved by irradiating the particles with

UV pulses from two xenon sources at ~280 nm and ~370 nm. Those are the optimal wavelengths in which typical bio-fluorophores are excited (e.g. tryptophan and nicotinamide adenine dinucleotide; Pöhlker et al., 2013). The fluorescence signals are detected using two photomultipliers equipped with bandpass filters to detect signals in the wavebands from 310 – 400 nm and 420 - 650 nm. The combination of two excitation wavelength and two detection wavebands provides 3 fluorescence channels (the 4[th] channel cannot be used due to interference from excitation). Channel 1 and 2 are both excited by the 280 nm xenon source and detected at 310 –400 nm and at 420 – 650 nm, respectively. Channel 3 is excited by the 370 nm xenon source and detected at 420 – 650 nm. In this study, the focus is placed on the total concentration of fluorescent particles, which fluoresce in at least one of all three channels, and on the fluorescent biological aerosol particle (FBAP) concentration. FBAP is thereby identified based on the criterion from Toprak and Schnaiter (2013), and refers to particles which simultaneously fluoresce in both channel 1 and 3, or which simultaneously fluoresce in channel 1, 2, and 3. It should be pointed out that fluorescence in any of the 3 channels can be impacted by non-biological particles such as dust; at the same time, using the criterion from Toprak and Schnaiter (2013) to determine FBAP can lead to an underestimation of biological particles (Savage et al., 2017). Moreover, a particle is considered fluorescent in a certain channel, if the fluorescence signal exceeds the threshold set for this channel. Typically, the threshold for each channel is obtained using the mean + three times the standard deviation of force trigger background measurements. However, during the campaign, no forced-trigger background measurements were conducted, hence, the fluorescence threshold was inferred from forced-trigger data acquired after the INUIT/CLACE campaign during free-tropospheric conditions.

Moreover, the concentration of equivalent black carbon (eBC) was measured with an aethalometer (model AE31, MAGEE scientific, Berkeley, California, USA). The eBC is calculated from the attenuation coefficient measurements by applying the factory standard mass attenuation cross sect. of 16.6 $m^2$ $g^{-1}$ at 880 nm. At this wavelength, eBC is the primary absorber such that an impact from mineral dust and brown carbon can be neglected.

### 2.5 Ice particle residual (IPR) analysis during in-cloud conditions

Small and thus freshly formed ambient ice particles were sampled using the ice-selective inlet (Ice-CVI; Mertes et al., 2007) to analyse the residuals of the ice particles when the sampling site was in-cloud. Briefly, by the geometry of the inlet head, only particles < 50 µm enter the inlet. A further reduction of sampled particle sizes (< 20 µm) is achieved by a virtual impactor. Since supercooled cloud droplets are still within this size range, they are impacted on two cold plates where they freeze upon collision, while the ice particles will bounce off. Any non-activated aerosol particles are excluded by following the streamlines of a downstream counterflow virtual impactor. This allows the selection of ice particles only, which are in a size range between 5 µm (lower particle cut off diameter of the counterflow virtual impactor) and 20 µm (upper particle cut off diameter of the virtual impactor). The ice particles are then sublimated such that the IPR can be further analyzed. Here, the IPR are investigated for their size using an Ultra-High Sensitivity Aerosol Spectrometer (UHSAS; Droplet Measurement Techniques, Longmont, USA), an OPS (0.3 – 10 µm; TSI 3300, TSI Inc. Shoreview, USA), and an optical particle counter (OPC, Grimm Aerosoltechnik, Freilassing, Germany), which are all optical-scattering, laser-based aerosol particle spectrometer systems. Moreover, their chemical composition was analyzed using the ALABAMA. For different analysis methods (scanning electron microscopy) we refer the reader to Eriksen Hammer et al. (2018).

### 2.6 Particle transport model and source sensitivities

Source emission sensitivities are derived with the Lagrangian Dispersion model FLEXPART (FLEXible PARTicle dispersion model; Stohl et al., 2005; Sturm et al., 2013; Pandey Deolal et al., 2014) which has a high grid resolution of 0.2° × 0.2° in the Alpine region, in combination with the numerical weather prediction model COSMO (Consortium for small-scale modeling). Every 3 hours, a total of 50,000 particles are released from JFJ and tracked backward in time to determine contact with the boundary layer. The calculations are driven by ECMWF IFS (European Centre for Medium Range Wheather Forecasting

Integrating Forecasting System) windfields. Assumptions for the FLEXPART simulations are: 1) 3000 m above sea level was assumed to be the ideal starting altitude for the backwards simulations of the aerosol particles sampled at JFJ, after tests with CO as a tracer for the current model configuration (Brunner et al., 2012). 2) For the determination of the source emission sensitivities only particles that have contact with the first 100 m above ground are considered. This is also the minimum mixing layer height in the model. 3) For the simulations, particles were tracked for ten days backward in time. 4) Furthermore, these are normalized residence times, which were divided by the air density (approx. factor 1.2 at sea level). To compare the air mass origin calculated with FLEXPART with the measured chemical composition of the aerosol particles, we consider only clearly definable source regions, such as marine surface areas and desert areas. Marine surface areas in this study consist of the following oceans: The Atlantic Ocean north of the equator, the Mediterranean Sea, the Black Sea, the Baltic Sea, the North Polar Sea and the North Sea. For the selection of the desert areas, a country-specific selection was made. In this study, the countries making up North Africa (from the Mediterranean to the Sahel) and the countries of the Middle East are taken together. All those countries are listed in Tab. S2 (Supplement) and are assumed to have at least partially semi-arid or arid areas that are potential sources of mineral dust particles.

Moreover, to improve the understanding of travel times from the marine boundary layer to JFJ, the FLEXPART simulations are tracked for the location, time, and intensity of the marine boundary contact for air masses sampled at JFJ. The residence time of model particles below 100 m above sea level is considered and their travel time after leaving the marine boundary layer is recorded.

In principle, the FLEXPART simulations for mineral dust and sea salt remain only qualitative, as the wind speeds at the potential source region of the particles are not considered. High residence times above source regions could therefore coincide with windless conditions resulting in little aerosol emission and thus cause a mismatch. However, short residence times could also be accompanied by high wind speeds above the source regions and high particle loading of the air, which could also result in a misinterpretation of the particle source. A further limitation of the FLEXPART simulations results from the non-consideration of deposition processes. This means that, for example, if air masses experience precipitation along the transport pathway to the measurement site, low aerosol particle concentrations can coincide with high residence times above source regions. This could be more pronounced for sea surfaces, since air masses coming from the sea are more humid and therefore there is a higher probability of precipitation than e.g. for dry Saharan air. However, the dry deposition of the particles is also decisive and depends not only on the meteorological conditions but also on the traveling time of the air mass from the source region to the measurement site. Despite such limitations, we consider the FLEXPART simulations to be a useful addition to chemical particle analysis to better identify potential source regions of INPs as has been shown in Boose et al. (2016a), Lacher et al. (2017), and Lacher et al. (2018a).

## 3. Results and Discussion

### 3.1 General description of measurement period

Observations of particle concentrations and $NO_y/CO$ ratios indicated that JFJ was for the vast majority of the sampling time within the free-troposphere during the field campaign (>90%; see Supplement Fig. S4). The ambient temperature ranged from -5 °C to -18 °C (Fig. S4, panel b), which is a relevant range in which MPCs can form. The prevailing local wind directions were South-East and North-West (Fig. S4, panel c), which reflects the dominating wind directions at the JFJ due to its location between the peaks of Mount Mönch and Mount Jungfrau. Wind velocities were below 20 m s$^{-1}$ (Fig. S4, panel c), which is the upper threshold for effective aerosol particle sampling via the GAW total inlet (Weingartner et al., 1999). The frequency of cloud presence (Fig. S4, panel e) is calculated using the difference of ambient temperature and sky temperature following the Stefan-Boltzmann law (Herrmann et al., 2015), and shows that the station was frequently exposed to in-cloud conditions.

Online measurements of [INP]$_{-31}$ with HINC show a fluctuation of several orders of magnitudes (Fig. 2, panel a). Two periods with elevated [INP]$_{-31}$ are identified, from January 22 – 27 and from February 11 – 17. Those periods occur simultaneously with an increase in particle concentrations > 0.5 µm (Fig. 2, panel a and Supplement Fig. S5). In addition to the online INP

measurements, filter-based [INP] were determined with FRIDGE and INSEKT in the temperatures between -20 °C and -30 °C (FRIDGE), and between -10 °C and -20°C (INSEKT), providing an insight into [INP] at higher temperatures (Fig. 3). Those measurements show that during the period January 22 – 27, February 11 – 17, [INP]$_{-30/-25}$ are elevated while [INP]$_{>-20}$ do not show higher concentrations, which reveals that the elevated INP population is only ice active below -20 °C. A common and important INP type in this temperature range is mineral dust (Hoose and Möhler, 2012; Kanji et al., 2017; Murray et al., 2012),

which might have dominated the INP population and would be supported by the increase in large particle fraction (Fig. 2, panel b; Fig. S5), also supported by Fig. 6 and related discussion in section 3.3.7). Indeed, it was found by Kammermann et al. (2010), that the concentration of supermicron particles is indicative of dust influence at JFJ. Moreover, as depicted in Fig. 3, the different INP instruments show a general good agreement at similar measurement temperatures (-20 °C for INSEKT and FRIDGE; -30°C for FRIDGE and HINC), which supports a temperature-dependent INP variability rather than an instrument-

specific variability.

Figure 2, panel c, also shows the relative abundance of particle types (by number) measured by the LAAPTOF. The corresponding representative mass spectra and the size resolved number fraction can be found in the Supplement, Fig. S10. ALABAMA data are not included in Fig. 2, because it started to measure with some delay, and it was connected to the CVI inlet during most of the cloud times. Figure 2, panel d, shows the fractional mass contribution of the non-refractory compounds

measured by the C-ToF-AMS. Sulfate and organics dominate the mass fractions, with occasional episodes of high nitrate and ammonium. Some general features of the chemical composition are observed in both the LAAPTOF and the C-ToF-AMS results. For example, times with high sulfate mass fraction measured by the C-ToF-AMS coincide with high number fractions of the particle types "secondary inorganics (NH4, NO3, SO4)", "K, organics, sulfate", and "more mixed/aged" measured by the LAAPTOF. Times with high organic mass fraction coincide with high number fractions of the "SOA and phosphate" and

"K, org, sulfate" types. Figure 2, panel b (right axis), also shows the total non-refractory submicron mass concentration measured by the C-ToF-AMS. The measured mass concentration was low, typically below 1 µg m$^{-3}$. The time series of the non-refractory mass concentration follows closely the number concentration for particles larger than 0.1 µm (panel b, left axis), indicating that the mass concentration is dominated by accumulation mode particles larger than 0.1 µm. No general dependence of particle composition or number concentration on the wind direction (Fig. 2, panel e) is observed.

**3.2 Source region analysis using FLEXPART**

In order to identify potential source regions for INPs, we performed a particle transport analysis for times during which elevated (above median values) $n_s$ and [INP]$_{-31}$ were measured ($n_s > 8.19 \times 10^8$ m$^{-2}$ and [INP]$_{-31} > 2.1$ stdL$^{-1}$, as indicated by the dashed lines in Fig. 2, panel a). Those median values are determined from a multi-year and multi-season analysis for free-tropospheric conditions at JFJ, using the same INP instrument (Lacher et al., 2018a). This analysis reveals that potential source regions

during times when elevated $n_s$ values were measured are the Atlantic Ocean and part of the Saharan desert (Fig. 4, panel a). During times of high [INP]$_{-31}$, the air mass origins were traced back to be the Saharan desert, the Mediterranean Sea, South-East/East Europe and the Middle East (Fig. 4, panel b). This is a first-order assessment of potential sources regions for INP, and in the following we will investigate the relationship between INP and aerosol particle chemistry.

**3.3 Single particle ions correlating with [INP] and $n_s$**

The time series of all ions up to m/z 250 of both polarities, measured with the LAAPTOF and the ALABAMA, were correlated to the time series of [INP]$_{-31}$ and $n_s$. This results in a total of 16 data sets (two single particle instruments, two correlation methods (see section 2.3.2), two polarities, two variables ($n_s$ and [INP]$_{-31}$) (Tab. 1, and Figs. S7 and S8 in the Supplement). In

the following, we present the results of those correlations for the possible particle types inferred from assigning ions to the observed m/z values. A selection of possible ions for each meaningful correlator m/z value listed in Tab. 1 and the assignment to possible particle types can be found in Tab. S1. The interpretation of particle components and particles types was achieved by the comparison with existing reference mass spectra from both mass spectrometers (see Figs. S9 and S10, panel a), as well as a m/z-to-m/z correlation analysis. The latter method provides information about which ions show a similar time series and thus can either represent isotopes of one element or different molecular fragments of the same original substance. Finally, single meaningful correlators were only assigned to a particle type if other meaningful correlators also indicated the same particle type and if this could be confirmed by both single particle mass spectrometers. When assigning ions to m/z values, it must be taken into account that different ions can be assigned to an integer m/z value, which in turn means that a single m/z value can be assigned to several particle types. This may result, for example, in two different ions of the same m/z value having increased correlations with the INP variables and thus appearing for the same polarity and m/z value for different particle types. Moreover, we also investigate ions with negative correlation coefficients. Furthermore, it should be noted that several particle types can be mixed internally due to long range transport. Therefore, it would not be surprising to find the marker ions in almost all the particle types. At the end of this section, we also discuss the differences between the two single particle mass spectrometers, and the correlation methods.

### 3.3.1 Sea spray

In the analysis of both instruments, we find that sea spray related ions have elevated correlation coefficients with both $[INP]_{-31}$ and $n_s$ (Tab. 1, Tab. S1). The chlorine anions $^{35}Cl^-$ and $^{37}Cl^-$ as well as cations with m/z 46 ($Na_2^+$), 81 ($Na_2^{35}Cl^+$), 83 ($Na_2^{37}Cl^+$) in the LAAPTOF (Tab. 1, panel a) and $^{35}Cl^-$, $^{37}Cl^-$, 23 ($Na^+$), 62 ($Na_2O^+$), 63 ($Na_2OH^+$), 78 ($K_2^+$), 12 ($Mg^{2+}$), 108 ($Na_2NO_3^+$), 181 ($KNa_2SO_4^+$) and 165 ($Na_3SO_4^+$), in the ALABAMA show positive correlation coefficients ($r$) between 0.31 and 0.57 (Tab. 1, panel b). Whether these correlations indicate that sea spray particles act as INP or that sea spray particles arrive in air masses together with other particles that act as INP remains open and cannot be verified by this correlation analysis. Particles of marine origin can be ice-active (e.g. Wilson et al., 2015), which is further discussed below by including data on air mass origin (see section 3.3.7).

### 3.3.2 Mineral dust

Mineral dust (Tab. 1, Tab. S1) is certainly a particle type that is expected to act as an INP. In general, we find correlation coefficients between ions indicative of mineral dust with both $[INP]_{-31}$ and $n_s$ in the range of 0.32 to 0.60. For example, the cations m/z 44 ($SiO^+$), 56 ($CaO^+$), 57 ($CaOH^+$), 75 ($CaCl^+$) as well as the anions m/z 60 ($SiO_2^-$), 76 ($SiO_3^-$), and 77 ($HSiO_3^-$) appear in the LAAPTOF correlation table with correlation coefficients between 0.3 and 0.6, in the ALABAMA data set we find cations such as m/z 7 ($Li^+$), 12 ($Mg^{2+}$), 23 ($Na^+$), 24 ($Mg^+$), 39 ($K^+$), 41 ($K^+$), 40 ($Ca^+$), 133 ($Cs^+$), 138 ($Ba^+$) and anions with m/z 43 ($AlO^+$), 76 and 77 ($SiO_3^-$ and $HSiO_3^-$) on the list with correlations coefficients above the threshold. Differences between both instruments can be explained by the laser energy differences used for ablation and ionization and are discussed in more detail below (section 3.3.5). Moreover, the instruments are not covering the same sampling time, which can also be an explanation for this finding.

### 3.3.3 Elemental carbon (EC)

Another interesting group of ions are the pure carbon ions $C_n^+$ or $C_n^-$ (Tab. 1, Tab. S1). The number of carbon atoms ($n$) reaches up to 12 for ALABAMA cations (m/z 144 = $C_{12}^+$), but for anions in the ALABAMA only $C_1^-$ (m/z 12) is found in the best-correlating ion analysis (Tab. 1, panel b), with correlation coefficients between 0.34 and 0.62. However, it should be mentioned that $C_1^-$ (m/z 12) in particular is not a unique feature for elemental carbon, but is also frequently observed in mass spectra of other particle types. In the LAAPTOF data set, higher carbon atom numbers (up to $C_6^-$) are found for the anions (r = 0.32 – 0.47) but no cations were detected (Tab. 1, panel a). Pure carbon ions $C_n^+$ signals can also be contributed by organics, e.g., in

LAAPTOF, SOA can generate very intensive $C^+$. However, our finding shows that the other organic ions, containing also hydrogen or oxygen atoms, are almost absent in the results with positive correlation coefficients. This indicates that the pure carbon ions $C_n^+$ signals are mainly from EC. Nevertheless, a few exceptions exist which are 59 ($C_2H_3O_2^-$) and 73 ($C_3H_5O_2^-$) as well as 42 ($CNO^-$), 58 ($C_3H_8N^+$), and 76 ($C_3H_{10}NO^+$), the latter indicating biological material or amines, but the pure carbon signals are dominant. The increased correlation of particles showing a typical ionic pattern of elemental carbon (see Fig. S9, panel a, presumably soot or other combustion-related particles) with [INP]$_{-31}$ and $n_s$ might indicate a possible connection to heterogeneous ice nucleation. In addition, [INP]$_{-31}$ also has a correlation coefficient of 0.5 with eBC, which is slightly higher compared to correlation coefficients for meteorological parameters and aerosol particle concentration < 0.1 µm (Fig. S6). We note that this does not necessarily translate to elemental carbon containing particles being ice nucleation active, but possibly other species transported in the same air mass could be the INPs (see also section 3.3.7). Laboratory studies of BC particles produced from synthetic fossil fuel are shown to be poor INP in the immersion freezing mode for temperatures above -38 °C (Mahrt et al., 2018; Kanji et al., 2020, Friedman et al., 2011; Chou et al., 2013), which question the role of pure fossil fuel BC on ice nucleation in MPCs. Recent studies indicate the regional importance of BC particles from biomass burning events (Schill et al., 2020), and are related to mineral phases of the biomass burning particles (Jahn et al., 2020). Interestingly, BC being a tracer with appreciable correlation with [INP] has also been reported for measurements conducted in the winter in the Boreal forest at Hyytiäla (Paramonov 2020). Thus, our findings do not necessarily suggest that EC containing particles are ice nucleation active, but potentially other internally or externally mixed particles could be the INPs, revealing that there was an anthropogenic impact on ice nucleation during this observation.

### 3.3.4 Ions with negative correlation coefficients

Although $r^2$ was chosen to determine which ions are meaningfully correlated with [INP]$_{-31}$ and $n_s$, we consider $r$ to also identify negative correlations. Here, we find two organic cations in the LAAPTOF data set (m/z 13, $CH^+$ and m/z 29, $C_2H_5^+$), but several sulfate-related anions in the ALABAMA data: m/z 96 ($SO_4^-$), m/z 97 ($HSO_4^-$), 99 ($H^{34}SO_4^-$), 141 ($CHO_2SO_4^-$ or $C_2H_5OSO_4^-$), 155 ($C_2H_3O_2SO_4^-$ or $C_3H_7OSO_4^-$), 177 ($HSO_4SO_3^-$, $CH_3SO_3H_2SO_3^-$), 195 ($H_2SO_4HSO_4^-$), 217 ($MgH(SO_4)_2^-$, $Na(HSO_4)_2^-$ or $NH_4NaSO_4SO_3^-$), and 233 ($KSO_4H_2SO_4^-$). Some of the listed ion identifications are tentative and have been reported in the literature (e.g. Froyd et al., 2010), but all represent possible sulfate-containing fragments are of inorganic or organic sulfate compounds. Such compounds are typically water soluble and are therefore not expected to contribute to immersion freezing INPs (e.g. Pruppacher and Klett 1997; Cantrell and Heymsfield, 2005; Kanji et al., 2017) at conditions assessed in HINC. Moreover, we do not expect that those components reduced the ice nucleation ability of the ambient aerosols by a coating effect in the immersion freezing regime (e.g., Kulkarni et al., 2014; Kanji et al., 2019).

### 3.3.5 Differences between the mass spectrometers

In most of the cases we find agreement between the two mass spectrometers ions that showed a relationship to [INP]$_{-31}$ and $n_s$. There are two noteworthy exceptions: a) mineral dust related ions and b) sulfate-containing ions. There are more mineral dust related ions clearly correlated with [INP]$_{-31}$ and $n_s$ in the LAAPTOF data set than in the ALABAMA data set. A likely reason for this is the different particle size and particle type dependent detection efficiency of the two instruments (cf. section 2.3.1). As mentioned before, LAAPTOF employs a higher power density than ALABAMA resulting in a higher degree of fragmentation in the mass spectra of LAAPTOF. These smaller organic fragments interfere less with mineral dust related fragments. Thus, interferences between organic fragments and mineral dust-related fragments, e.g., m/z 40 ($Ca^+$ or $C_3H_4^+$), 44 ($SiO_2$, $C_3H_8^+$ or $CO_2^+$), and 57 ($CaOH^+$, $C_4H_9+$, $C_2O_2H^+$, $C_2H_5CO^+$, $C_3H_5O^+$) are more likely to occur in the ALABAMA mass spectra, as these organic ions are composed of longer chains of atoms and are thus more likely to undergo fragmentation in the ablation and ionization process. The time series of the mineral dust related ions detected in the ALABAMA can therefore be superimposed by the organic ions. As our data shows almost no correlation between organic ions and [INP]$_{-31}$ or $n_s$, this explains the finding that the apparent correlation between mineral dust related ion and [INP]$_{-31}$ or $n_s$ is weaker in the

ALABAMA than the LAAPTOF data set. Possible reasons for differences in detecting sulfate-containing ions could be the higher power density of the LAAPTOF, which results in smaller fragments, thus it is weaker in identifying organosulfates, e.g., m/z 141 ($CHO_2SO_4^-$ or $C_2H_5OSO_4^-$) and 155 ($C_2H_3O_2SO_4^-$ or $C_3H_7OSO_4^-$), and large inorganic sulfate ions, e.g., 195 ($H_2SO_4HSO_4^-$), 217 ($MgH(SO_4)_2^-$, $Na(HSO_4)_2^-$ or $NH_4NaSO_4SO_3^-$), and 233 ($KSO_4H_2SO_4^-$), than the ALABAMA.

### 3.3.6 Differences between the correlation methods (method 1, method 2, [INP]$_{-31}$, and $n_s$)

In general, we find that correlations between time series of ions and [INP]$_{-31}$ are higher than for $n_s$ (Tab. 1, Figs. S7 and S8). This does not imply that ice activation is not controlled by the aerosol particle surface, but could imply that the particle composition is playing an important role. The calculation methods can also contribute to this: $n_s$ is derived by normalising [INP]$_{-31}$ to the total available aerosol surface area (see Eq. 1). Thus, at high aerosol load it can be that $n_s$ will decrease (or stays constant) while [INP]$_{-31}$ increases only marginally compared to the total aerosol surface, thus potentially leading to a better correlation between the ions and [INP]$_{-31}$. On the other hand, both methods 1 and 2, that were used to calculate the time series of ions, are normalized methods: Method 1 is normalized to the total number of recorded single particle mass spectra in the time interval (20 minutes). Method 2 is normalized to the total signal height of the spectra. Thus, both methods are not sensitive to the total aerosol particle number, surface, or mass concentration in the way that is used in the calculation of $n_s$. Further, we find that the highest correlation coefficients between the time series occurs using method 2 and [INP]$_{-31}$. The ion fraction in method 2 represents the average composition of all particles during the 20-minute time interval, whereas method 1 reports the fraction of particles that contain a certain ion. The higher correlation coefficient found between [INP]$_{-31}$ and averaged chemical compositions, as compared to the fraction of single particle ions present in the 20-minute time interval reported by method 1, suggests that not only the presence of a substance causes ice activation, but that the relative amount of the substance in the averaged particle ensemble is important to the ice activation properties of the particle ensemble. The higher correlation with [INP]$_{-31}$ as with $n_s$ supports our initial statement that a non-negligible influence of particle composition impact ice nucleation. Thus, not only aerosol physical properties such as size or surface area, which are typically dominant predictors of [INP] (e.g. DeMott et al., 2010), are important but also chemical composition.

### 3.3.7 Time series of best correlating ions

To illustrate the findings discussed above, we summed the time series of the best correlating ions of a specific particle type to create a proxy for a more general particle type. For this, we chose correlation coefficients with [INP]$_{-31}$ based on method 2, as this combination yielded in general the highest correlation coefficients. For example, for the sea spray time series we added the time series of m/z +23, -35, -37, +62, +63, +78, and +165 (ALABAMA). Similarly, we created a time series for sea spray also from LAAPTOF data, for mineral dust and for elemental carbon (ALABAMA and LAAPTOF), and for sulfate-containing particles (ALABAMA only) using the marker ions of the respective particle types listed in Tab. S1 (bold marked). The signals at m/z +12, +24, +39, +41 were not considered for the sea spray and mineral dust time series, because these are very common signals across all particle types. The ions at m/z +12 and +24 can be attributed to both carbon and magnesium. Although m/z +12 clearly shows an increased intensity in the EC type compared to the other particle types listed, it is less clear for m/z +24. Therefore, m/z +24 was not used as a marker for any of the particle types. The signals at m/z +39 and +41 are mainly indicative of potassium, which is a common component of mass spectra due to its ionization energy. For example, potassium is occurring in biomass burning particles (e.g., Silva et al., 1999) but is also present in sea water and in mineral dust. Thus, the ions m/z +39 and +41 were not considered here in the analysis.

These time series are displayed in Figs. 6 and 7, along with the [INP]$_{-31}$ time series and particle source region information. For the latter, we used emission footprint sensitivities derived from FLEXPART. We summarized the North African and Middle East countries to account for possible source regions of mineral dust and the marine surfaces to account for possible sea spray

particle sources (as explained in section 2.6). Figure 5 (panel a) shows that the time series of the footprint emission sensitivity calculation for dust surfaces, respectively for the North African and Middle East countries, are elevated during times when [INP]$_{-31}$ is high, and are accompanied by elevated mineral dust marker values, e.g. on January 23, 27, and 29, and February 3, 12 - 17, and 19 (Fig. 5, panel b). Moreover, the mineral dust proxies from both mass spectrometers agree well (Fig. 5, panel b). The correlation with the [INP]$_{-31}$ time series has already been discussed above and indicates that mineral dust particles transported from source regions in North Africa and the Middle East to JFJ act as INP at -31 °C.

The time series of the EC proxies from ALABAMA and LAAPTOF are shown in Fig. 5, panel c. Here, there is some discrepancy between the instruments, for example during the period between February 14 and 19. Then ALABAMA data show a higher contribution of EC than during the first part of the campaign (January 29 –31), whereas in the LAAPTOF data the EC contribution remains similar when comparing January 23 – February 7 and February 14 –19, which might be related to differences in the instruments (see section 3.3.5). Thus, the elevated [INP]$_{-31}$ during February 14 – 19 is occurring not only during times of elevated dust, but also elevated EC concentrations. As mentioned before, many studies have reported EC particles do not contribute to heterogeneous ice nucleation above -38 °C (e.g., Kupiszewski et al., 2016; Vergara-Temprado et al., 2018; Mahrt et al., 2018; Kanji et al., 2020). It is possible that the air masses transporting mineral dust from the North African and Middle East source regions also transport polluted air bringing EC containing particles along the transport pathway. This may result in the observed correlation between INP and EC containing particles without the latter being active as INP and is supported by an increase in the EC signal occurring simultaneously (ALABAMA, see Fig. 5 panels b, c) with increase in the dust markers during January 31 and February 14 - 18.

Figure 6 shows the timeseries of the relative contribution of marine surfaces from FLEXPART, together with [INP]$_{-31}$ (Fig. 6, panel a), and the proxies for aged sea spray particles from both the ALABAMA and LAAPTOF (Fig. 6, panel b). Both measurements from the single-particle mass spectrometers show similar trends, although the ALABAMA data show much less variation in the second part of the campaign (after February 11) than the LAAPTOF data set. The differences in the ALABAMA time series between the first and second half of the campaign are also partly due to changes in the instrument configurations (as described in section 2.3.2). However, the time series of the sea spray proxies from the mass spectra do not match the FLEXPART emission footprint sensitivities for open ocean and sea surfaces (Fig. 6, panel a). Also, the [INP]$_{-31}$ time series does not match the open water source regions from FLEXPART. Interestingly, the calculated air mass travel times from FLEXPART between the marine boundary layer and the measurement site reveal similarities with the sea spray proxies. For example, the period when the single-particle mass spectrometers recorded a stronger signal from sea spray particles (2 to 5 February) coincides with comparatively short travel times of the air masses to the JFJ (12 to 24 hours), mainly originating from the marine boundary layer over the western Mediterranean Sea (Fig. S11). The short travel time is in line with a reduced impact of sedimentation losses, wet removal, and mixing/dilution with other air masses of different origins. The FLEXPART analysis shows that travel times from the marine boundary layer to JFJ, can vary between approximately 24 hours and more than 96 hours. However, we should point out that the FLEXPART model is not able to separate coastal regions from open ocean for this analysis, such that the open water classification used may not be the best way to represent sea spray source regions which tend to occur along shorelines during wave breaking and as function of windspeed (e.g. Vergara-Temprado et al., 2017).

Apart from this, both the sea spray and the [INP]$_{-31}$ time series show a similar temporal evolution as the dust source region emission sensitivities (Fig. 5). A possible explanation for this might be found in the dry saline lake beds in the deserts have been identified as source regions for salt particles (Prospero et al., 2002). For example, Formenti et al. (2003), observed that airborne particles originating from the Sahara that were sampled off the North African coast were a mixture of aluminium-silicate based minerals and NaCl bearing salts. This may well explain our observed correlation between dust particles, NaCl containing particles and INP.

Figure 6 (panel c) shows the time series of the sulfate proxy ions detected by the ALABAMA. For sulfate ions, we observed a negative correlation with $[INP]_{-31}$ (see also Tab. 1). Especially in the first half of the campaign, it can be seen that the sulfate signal in the ALABAMA is high at times when $[INP]_{-31}$ is low (January 28 – 30, February 2 –3, February 7, and February 9). After February 11, when $[INP]_{-31}$ are higher along with the mineral dust signals and the emission sensitivities of the dust source regions, the sulfate ion time series shows only slight fluctuation and shows no similarities to the $[INP]_{-31}$ time series. Furthermore, no similarities with the dust source region emission sensitivities are observed. A comparison of the ALABAMA sulfate ion time series with the sulfate fraction measured with the AMS (Fig. 6, panel c, right panel) shows a good agreement in the time points of the variations, although not in the strength of the fluctuations. Thus, the timeseries of the ALABAMA and AMS sulfate measurements are anticorrelated to the $[INP]_{-31}$ time series which is not surprising given soluble aerosol particles are not expected to contribute to heterogeneous ice nucleation above -38 °C.

### 3.4 Fluorescent particles

During the duration of the campaign, a WIBS instrument measured fluorescent particles in a size range between 0.5 - 20 µm. Only a minor fraction of fluorescent particles was found to be FBAP (0.013) according to the criteria of Toprak and Schnaiter (2013). However, it should be noted that the application of the chosen criterion to identify FBAP (simultaneous fluorescence in channel 1 and 3, or in channel 1, 2 and 3, see section 2.4) can lead to an underestimation of biological particles (Savage et al., 2017). Thus, we also considered the total fluorescent particles concentration, with fluorescence in at least one of the 3 channels, for the correlation coefficient analysis. We find the ranked correlation coefficient between $[INP]_{-31}$ and FBAP to be low (0.16), while it is quite high for fluorescent particles (0.63; Fig. S6, Supplement). However, the correlation with the total particle concentration (sum of fluorescent and non-fluorescent particles) determined with the WIBS is even higher (0.75). As the WIBS measures only particles > 0.5 µm, the observed relationship between $[INP]_{-31}$ and fluorescent particle concentrations is confounded by a size-effect, as the large particle size fraction is enough to explain this good correlation. As depicted in Fig. S10, panel b, mineral dust particles mainly occur in the size ranges > 0.5 µm, and they can also show fluorescence. Thus, we cannot clearly verify a contribution from biological particles to INPs active at -31 °C during the INUIT/CLACE field campaign which is unsurprising, given the season, dominant snow cover and free troposphere conditions at the site.

Moreover, we would like to point out that the strong correlation between $[INP]_{-31}$ and the total particle concentration measured with the WIBS is the highest observed during this campaign, together with the particle concentration > 0.5 µm measured with the OPS (Fig. S6).

### 3.5 Ice particle residuals (IPR)

### 3.5.1 Physical cloud characteristics related to the number and size distribution of IPR

Cloud residuals of 16 separate cloud events were sampled by the Ice-CVI, sub-divided into 43 cloud periods by the shape of the residual particle size distribution. This was done on the basis of the observed concentration ratio $R$ of the size range 300 - 600 nm to 80 - 100 nm:

$$R = \frac{N_{300-600}}{N_{80-100}} \tag{2}$$

For $R$, three types of cloud periods were defined: $R < 1$, $1 < R < 3$ and $R > 3$. $R > 3$ denotes cloud periods where the IPR number size distribution shows a clear minimum or at least a strong decrease at around 100 nm (Fig. S12), whereas $R < 1$ denotes cloud periods where the relative amount of small IPR is very high compared to larger IPR sizes (Fig. S13). The cloud period type $3 > R > 1$ is in between and thus rather inconclusive and therefore not included in the further evaluation. Considering sufficient counting statistics based on the UHSAS and the ALABAMA measurements, 20 cloud periods remained for the condition $R > 3$ and $R < 1$ (Tabs. S3, S4 and S5) corresponding to 12 separate cloud events.

Since it is very unlikely to find INPs at a diameter below 100 nm (e.g. Vali, 1966; Pruppacher and Klett, 1997; Richardson et al., 2007; Cornwell et al., 2019), this feature of small IPR is attributed to the presence of secondary ice in the examined clouds. Therefore, the cloud periods presented in Fig. S12 / Tab. S3 and Fig. S13 / Tab. S4 are interpreted as MPCs with less and more impact of secondary ice production, respectively. This assumption is further supported by the fact that on average the cloud periods interpreted as being stronger influenced by secondary ice contribution have significantly higher IPR number concentrations. Such an implication small cloud residual sizes (< 200 nm) on secondary ice formation is also reported for measurements conducted at the Zeppelin observatory (Ny Ålesund) in the Arctic (Karlsson et al., 2021). During these periods the IPR concentration is also higher in the IPR diameter range larger than 300 nm. This implies that residuals from secondary ice particles can also contribute to the size range of IPRs expected from primary ice particles. Figure 7, panel a, shows the mean IPR number size distribution for both cloud categories averaged with respect to the duration of the single cloud periods.

### 3.5.2    Chemical composition of IPR

The IPR mass spectra recorded with the ALABAMA downstream of the Ice-CVI were analyzed separately for cloud periods with lower and higher contributions of small IPR as defined in section Sect. 3.5.1 (see Tab. S3 and Tab. S4). During the 20 analyzed MPC periods, a total of 640 IPR mass spectra were recorded with the ALABAMA. These spectra were clustered using a fuzzy c-means algorithm (Roth et al., 2016; Hinz et al., 1999; Bezdek et al., 1984), and the resulting clusters were assigned to particle types based on marker ions and reference particle mass spectra, resulting in eight different particle types. Figure 7, panels a and b, shows the IPR size distribution for cases of lower (Tab. S3; $R > 3$) and higher (listed in Table S4; $R < 1$) contribution of small IPRs, respectively. The particle number fractions of those two cloud conditions are depicted in Fig. 7, panel c, and the size resolved particle number fraction of the two cloud conditions are shown in Fig. 8. The IPR particle types will be discussed in the following. Additionally, a temporal evolution of the particle composition during a selected cloud event is shown in the Supplement (Figs. S14 - S16).

*Mineral dust and biological particles*: In agreement with previous IPR studies conducted at the JFJ station (Kamphus et al., 2010; Worringen et al., 2015; Schmidt et al., 2017; Eriksen Hammer et al., 2018), IPR of mineral origin were detected in almost every cloud event and were the most frequently detected IPR type by number (58%, also including Al type particles, see discussion below about "Aluminium and other metal containing particles"), even though ambient temperatures during the campaign were relatively high (> -17 °C). Activation of ice by mineral dust at such relatively high temperatures might be caused by biological material on the surface of dust particles, because many IPR contained ions indicating biological material (see Fig. S17). Another possible explanation for this relative high activation temperature could be that the mineral dust particles were highly ice-active dust particles, such as potassium-rich feldspars, which generally nucleate ice more efficiently and at higher temperatures as other mineral dust particles (Harrison et al., 2016). Moreover, the fraction of mineral dust containing IPR does not appear to be related to the proportion of small IPR, as it is found in both cloud period classifications. Moreover, mineral dust IPRs were detected in size ranges between 250 nm and 1500 nm (Fig. 8). Interestingly, the majority of dust IPR are submicron in size Particles dominated by ion signals indicating biological particles were also consistently found in both cloud period classifications, but to a general smaller proportion (< 10%), which was also reported by Schmidt et al. (2017) and Kupiszewski et al. (2015).

*Aluminium and other metal containing particles*: The aluminium particle type fraction is the second largest particle fraction, contributing more than 25% by number during cloud periods with a smaller contribution from small IPR and more than 15% during periods of high contribution from small IPR. Although aluminium ions in single particle mass spectra can indicate mineral dust, Eriksen Hammer et al. (2018) identified a subset of IPR containing aluminium as possible contamination from the sampling setup, despite all surfaces of the Ice-CVI were coated with a nickel alloy prior to the 2017 campaign. However, in this work the aluminium particle type detected by the ALABAMA contains additional ion signals of $Na^+$, $Cl^-$, cyanide ($CN^-$) and cyanate ($CNO^-$), thus we regard them as real ambient particles and count them as mineral dust particles (see discussion

above) rather than sampling artifacts. Such aluminium-dominated particles have also been observed in IPR by Schmidt et al. (2017) using the ALABAMA and by Kamphus et al. (2010) using the aerosol time-of-flight mass spectrometer (ATOFMS, TSI Model 3800) in combination with the Ice-CVI at the JFJ in winter 2013 and in winter 2007, respectively. Other metallic particles were detected with a similar fraction (below 10%) during times of low and high small IPRs, indicating that metal-containing particle act as INP, which agrees with previous findings (Kamphus et al., 2010; Eriksen Hammer et al., 2018; Schmidt et al., 2018).

*Sodium chloride particles*: Another frequently observed particle type is dominated by NaCl, indicating sea spray particles (e.g., Sierau et al., 2014). It should be mentioned that the described correlation between sea spray particles with INP (see Section 3.3.1) refers to total sea spray. In the following we distinguish between aged sea spray (derived from ions such as $Na_2O^+$, $Na_2O(H)^+$, $Na_2NO_3^+$, and $Na_3SO_4^+$, $NaSO_4^-$; Gard et al., 1998; Sierau et. al, 2014) and pure NaCl mass spectra (containing only $Na^+$, $Cl^-$, and various NaCl compounds such as $Na_2Cl^+$, $Na_3Cl_2^+$, $NaCl_2^-$). The main difference between the IPR composition during the two cloud period classes is the fraction of the pure NaCl particle type which is clearly higher (about 20%) during the cloud periods with a high concentration of small IPR ($R < 1$). This indicates that these pure NaCl particles do not originate from freshly formed small ice crystals as INPs, but result from sampling ice crystals formed by secondary ice production. The same effect may explain the previously reported observations of NaCl particles in IPR (Kamphus et al., 2010; Eriksen Hammer et al., 2018). This interpretation is further discussed in Section 3.5.3.

*Size resolved chemical fraction and internal mixtures*: The size distribution of the IPRs reveal a dominant submicron fraction (Fig. 8). This challenges the assumption that most of the INPs are supermicron in size (e.g. Mason et al., 2016), and might be related to the sampling location in the free troposphere, where particles typically undergo long-range transport and sedimentation processes prior to being sampled. The size-resolved examination of the particle fractions (see Fig. 8) shows that IPR containing mineral dust were detected in all size channels between about 150 nm and 3000 nm (vacuum aerodynamic diameter). IPR with biological and elemental carbon ion signals are mainly found in the size range between about 150 nm and 600 nm, whereas the pure NaCl particles have the highest contribution between about 400 and 1000 nm. We note that the attribution of the IPR spectra to particles types is based on the dominant ion signals of the mass spectra and is not an absolute assignment to a pure particle type. Thus, almost all IPR types also show traces of internal mixtures. For example, over 90% of the IPR contain signals of sodium ($Na^+$), over 80% contain signals of potassium ($K^+$) and over 70% of the IPR contain signals of $Cl^-$ and $CN^-$. Mineral dust particles in turn contain a variety of other signals that indicate sea spray, biological and biomass burning substances. Therefore, the IPR types cannot be considered completely independent of each other and represent internal mixtures.

### 3.5.3 Dependence of the IPR composition on the wind direction

At the JFJ, two main local wind directions prevail due to the topography around the station. To study the influence of wind direction on IPR composition, we selected air masses coming from either southeast or northwest to the JFJ station and sorted the cloud periods of Tab. S4 (those with high contribution of small IPR, $R < 1$, indicating more impact from secondary ice formation) for these two main flow directions (Fig. 9). Figure 9, panels a and b, shows the size distribution of the IPR for the two dominant wind directions, and panel c depicts the IPR composition. The fraction of the pure NaCl IPR type is significantly higher ($> 40\%$) for the north-westerly flow than for the south-easterly flow (about 3%), whereas the other IPR types show only slight differences between the two cases. Pure NaCl particles are generally considered poor INPs and freeze homogeneously or heterogeneously at temperatures below -38 °C (Kanji et al., 2017 and references therein) and thus should not be expected to occur in primary IPRs under the temperature conditions at the JFJ. In case the ice crystals would have originated from homogeneous freezing at higher altitude and settled to JFJ, they would have grown to sizes too large to be sampled with the Ice-CVI, which only collects freshly (small) nucleated ice crystals. We can also rule out that the observation of pure NaCl particles in the IPR population is due to collected cloud droplets, based on the working principle of the Ice-CVI, which impacts

supercooled cloud droplets on a cold plate where they freeze upon collision. In case that the Ice-CVI would also transmit liquid cloud droplets, one would also expect an enhanced abundance of sulfate and nitrate signals, which were detected in cloud droplet residuals (Roth et al., 2016; Kamphus et al., 2010). However, this was not the case in our work. Further, scavenging is unlikely, because firstly, the Ice-CVI samples only small, freshly nucleated ice crystals that have been growing only by diffusion (5 – 20 µm; Mertes et al., 2007), and secondly, scavenging would also apply to sulfate, nitrate, and organic particles that dominate the aerosol population at JFJ (see Fig. 2).

Thus, we conclude that the pure NaCl IPR type is associated with secondary ice processes. At the JFJ, the different orographic terrain profiles northwest and southeast of the station lead to stronger vertical wind speeds for the steep rise on the northern side of the research station, resulting in high peak supersaturations (Hammer et al., 2014). This is expected to favor the occurrence of persistent MPCs, as compared to the gentler slope on the south of the measuring station (Lohmann et al., 2016). These stronger updrafts at the northwesterly slope can transport larger droplets to higher regions with lower temperatures, which in turn increases the formation of secondary ice crystals (Lohmann et al., 2016; Korolev and Leisner, 2020). Various secondary ice processes, such as ice fragmentation due to surface and spicule bubble bursting during thermal shock, fragmentation during ice-ice collision or splintering during riming (Hallett-Mossop process), are favored by freezing of large droplets, although Hallett-Mossop has its maximum secondary ice production rate at higher temperatures than those occurring most of the time during our measurements (Hallett and Mossop, 1974; Korolev and Leisner, 2020). Furthermore, blowing snow or fluxes of surface hoar frost are also possible sources of secondary ice (Vali et al., 2012, Farrington et al., 2016, Lloyd et al., 2015; Beck et al., 2018). However, no correlation was observed between horizontal wind speeds at the JFJ and IPR composition. We note that the sulfate fraction in the IPR composition during times of an increased influence from secondary ice crystal formation (during small IPR samples) is low as compared to NaCl, although one would expect that sulfates are also present in droplets which contribute to secondary ice crystals. This could be explained by the original CCN mass being distributed over many small ice fragments upon fragmentation of the ice crystal. Thus, the sampled IPR is smaller than the initial CCN. This shifts the IPR size distributions to smaller sizes, such that sulfate-containing IPR are possibly too small to be detected in the ALABAMA size range, and a preferential sampling of NaCl particles are detected. Indeed, the size distribution from ambient sea spray particles shows a maximum > 1 µm (Fig. S10, panel a) while the IPR NaCl particle under northwest wind conditions have a maximum < 1µm (Fig. 10, panel b). Fig. S10 also shows that the sulfate containing aerosol particles in the ambient air have a maximum at smaller sizes than NaCl-containing particles. Moreover, the faster updraft velocities lead to a shorter lifetime of the cloud droplets before freezing, such that aqueous-phase reactions that lead to aged sea spray particle signatures (e.g., $Na_2NO_3^+$ and $Na_3SO_4^+$, Gard et al., 1998) are less likely to occur compared to a slow updraft cloud formation situation, which supports the dominance of the small IPRs with pure NaCl signatures. Which process is ultimately responsible for the secondary ice formation with the simultaneous occurrence of pure NaCl IPR cannot be determined from our measurements.

## 4. Conclusions

In this study we investigate the potential chemical characteristics and sources of immersion freezing INPs at -31 °C at the High Altitude Research Station Jungfraujoch (JFJ) in winter 2017, by combining INP concentration measurements from the online instrument HINC with measurements from two single particle mass spectrometers, the ALABAMA and the LAAPTOF, and with FLEXPART source emission model calculations. We correlate the time series of individual ions from the two mass spectrometers with the time series of [INP]$_{-31}$ and $n_s$ to determine potential relationships. Such correlation analysis allows to include also small ion signals which still might represent chemical substances rather than whole particle types, such that fewer initial assumptions have to be made, allowing a cross-particle type approach. Based on our analyses, sodium-, calcium-, silicon- and chlorine-containing ions in particular showed increased correlation with [INP]$_{-31}$ and $n_s$. We concluded that these ions originate from substances that are essentially due to mineral dust and sea salt particles. From those results we find consistent

evidence for an impact of mineral dust and particles of marine origin on ice nucleation, which is also supported by particle transport simulations. Moreover, the investigation of the timeseries of [INP] at temperatures from -10°C to -30°C, as measured the offline methods INSEKT and FRIDGE, reveals that only [INP] below -20°C follow similar trends to those of [INP]$_{-31}$, which could be due to mineral dust particles dominating the INP population. The different INP methods thereby show a good agreement in INP concentrations at similar measurement temperatures.

While mineral dust is known to play a major role in ice nucleation (e.g. Hoose and Möhler, 2021; Murray et al., 2012; Kanji et al., 2017), marine INPs are mostly assumed to be present in remote marine environments (Vergara-Temprado et al., 2018; McCluskey et al., 2019). From this work it is evident that locations far away from the sampling site, like the Saharan desert and the oceans, can still have a substantial contribution to the INP population. We also observe correlations between INPs and elemental carbon related ion markers, which coincide with those of dust, and interpret this as a dust related correlation. This is supported by the conclusions in Kupiszewski et al., (2016) and Eriksen Hammer et al., (2018), who do not find an enrichment of BC in IPR, but contrasts the finding of Cozic et al. (2008). The discrepancy to the study by Cozic et al. (2008) might be caused by the different absorption-based methods, which was likely impacted by dust interference in Cozic et al. (2008).

Our findings of the INP population at JFJ in winter 2017 are supported by the measurement of ice particle residuals (IPRs). Although ice crystals in the cloud formed at markedly higher temperatures compared to the freezing temperature of the INP$_{-31}$, it is still apparent that the major contributors are mineral dust (58 % ±32 % of IPRs, including a aluminum-type particle) and sea spray indicating particles (17 % ±20 % of IPRs, referring to fresh and aged sea spray particle types), consistent with our conclusions from the correlations of ion signals from mass spectrometry and INP and $n_s$. In addition, biological, metallic and elemental carbon containing particles were also found to be present in the ice residuals to lesser extent. However, it must be pointed out that the attribution of all mass spectra to particle types is only based on the dominant ion signals of the mass spectra and is not an absolute assignment to a pure particle type. Most particles detected were internally mixed, which is not surprising giving the typical residence time (several days) of particles in the free troposphere, during which they can undergo aging processes. In addition, most of the IPRs were submicron size, which indicates that smaller particles are contributors to INPs at the given ambient temperatures in the free troposphere. Moreover, a possible influence of ice crystals formed by secondary production processes on the IPR population was believed to occur when cloud periods with increased small sizes (80 - 100 nm) and high IPR concentrations possibly related to secondary ice crystals in agreement with a recent study of small cloud residuals (< 200 nm) impacting secondary ice in arctic clouds (Karlsson et al., 2020). The investigations showed that in particular pure NaCl particles were observed more frequently as IPRs during such cloud periods with a simultaneous northwesterly inflow to the measurement site, which cannot be attributed to primary INPs at the prevailing temperatures.

Our study presents the physical and chemical properties of INPs and overall aerosol particles, using measurements from single particle mass spectrometry, aerosol mass spectrometry, an ice selective inlet and model simulation for source emission sensitivities to identify potential source regions of INPs and IPRs. At the same time, our findings highlight the need for further investigations for the sources of INPs on a longer time period and in different seasons and years (e.g. using a similar measurement setup), and to directly investigate the identity of INP with a different instrumental setup, as e.g. a coupled cloud chamber with single particle mass spectrometry, and with ice-selective inlets.

*Data availability:* The data presented in this publication will be made available at DOI 10.3929/ethz-b-000484207. Note by the authors: Data and DOI link will be activated for public access upon acceptance of publication.

**Author contributions:** LL, HCC, SM, JoS, XS, and ZAK conceived the study. LL lead the overall writing of the manuscript with specific contributions from HCC, JoS, SM and XS. The manuscript was reviewed and edited by HS and ZAK. JoS coordinated the field activities. LL performed and analyzed the online INP measurements. HCC, XS and JoS performed and analyzed the mass spectrometry measurements. SM operated the Ice-CVI and analyzed the IPR size distribution. SH and HCC

conducted and analyzed the FLEXPART model calculations. MS performed and analyzed the trace gas measurements. MG-B provided the aerosol particle size distributions, the aethalometer and WIBS measurements. AM analyzed the WIBS data. DW and JaS conducted and analyzed the offline INP measurements from FRIDGE. LL, XS, OM, KH and TS contributed to the filter collection and analysis of INSEKT. LL, HCC, XS, JoS, SM, HS and ZAK interpreted the data and all authors contributed to reviewing the manuscript. JoS and ZAK oversaw the project.

**Competing interests:** The authors declare no conflicts of interest.

**Acknowledgements**

We thank the International Foundation High Altitude Research Station Jungfraujoch and Gornergrat (HFSJG) for the support and the opportunity to perform measurements, with a special thanks to the custodians Maria and Urs Otz, and Joan and Martin Fischer. Meteorological data were provided by MeteoSwiss. We also acknowledge all the participants of the INUIT/CLACE 2017 campaign. Erik Herrmann and Nicolas Bukowiecki are acknowledged for operating various aerosol instruments during the campaign. We thank Oliver Eppers and Udo Kästner for their support during the campaign, as well as the KIT technical team.

**Financial support**

Larissa Lacher and Zamin A. Kanji received funding from the Global Atmospheric Watch, Switzerland (MeteoSwiss GAW-CH+ 2014-2017). Johannes Schneider and Hans-Christian Clemen were funded by the Deutsche Forschungsgemeinschaft (DFG, German Research Foundation) – 170852269, within the research unit INUIT (FOR 1525), and from the EU Horizon 2020 research and innovation programme (grant no. 654109, ACTRIS-2). Stephan Mertes, Jann Schrod, and Daniel Weber were funded by the DFG – 170852269 within the research unit INUIT (FOR 1525). Ottmar Möhler, Thea Schiebel and Kristina Höhler were partially funded by the Deutsche Forschungsgemeinschaft (DFG), project No. 170852269 (Research Unit INUIT, FOR 1525). Harald Saathoff and Xiaoli Shen received funding from the European Union's Horizon 2020 research and innovation programme as part of the ACTRIS-2 project under grant agreement No 654109. The aerosol measurements by PSI were performed with financial support from MeteoSwiss (GAW-CH 2014-2017) and from ACTRIS2 project (EU H2020-INFRAIA-2014-2015 under grant agreement no. 654109, and Swiss State Secretariat for Education, Research and Innovation under contract number 15.0159-1. The opinions expressed and arguments employed herein do not necessarily reflect the official views of the Swiss Government).

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

## 4    Figures

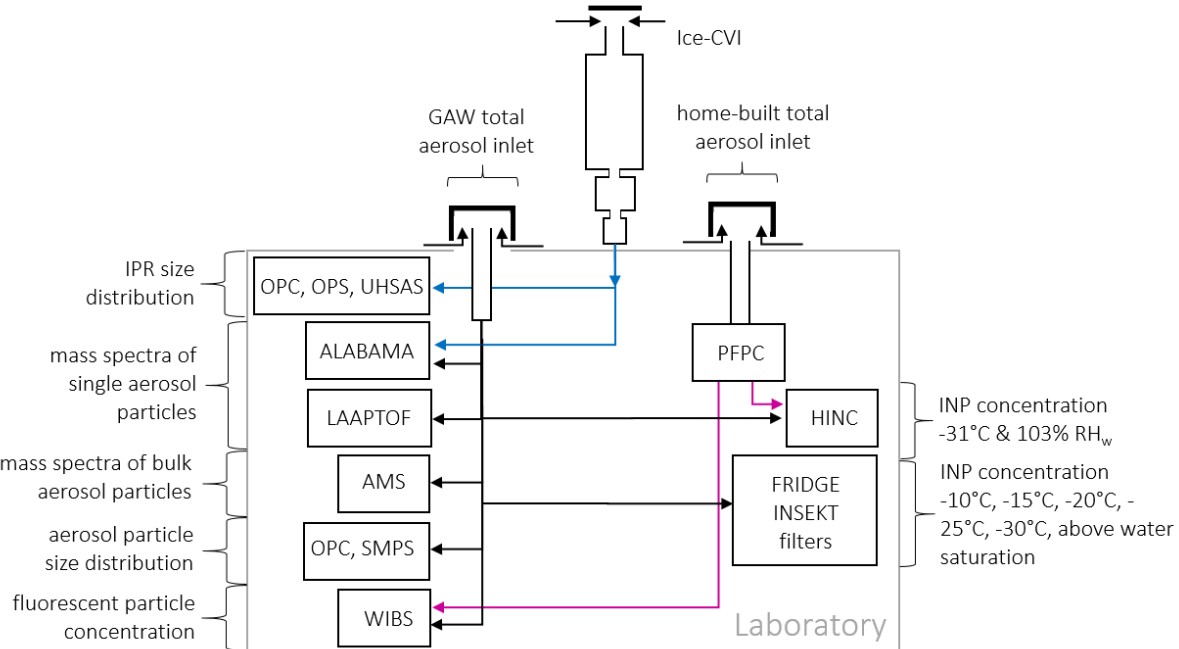

Figure 1: Setup of instruments at the Sphinx observatory at JFJ. The instruments were mostly sampling via the GAW total aerosol inlet (black line). The aerosol particle chemical composition was measured with ALABAMA,  LAAPTOF, and an

AMS; aerosol particle size was detected using an OPC and a SMPS; fluorescent biological particles were measured with a WIBS; online INP concentrations were measured with HINC and offline with INSEKT and FRIDGE. HINC and WIBS measurements were alternated at the GAW- and a home-built total aerosol inlet (black line) connected to the Portable Fine Particle Concentrator (magenta line). Measurements of IPR were performed at the Ice-CVI inlet (blue line) for IPR size (using an OPC, OPS, and an UHSAS) and for chemical composition (using ALABAMA).

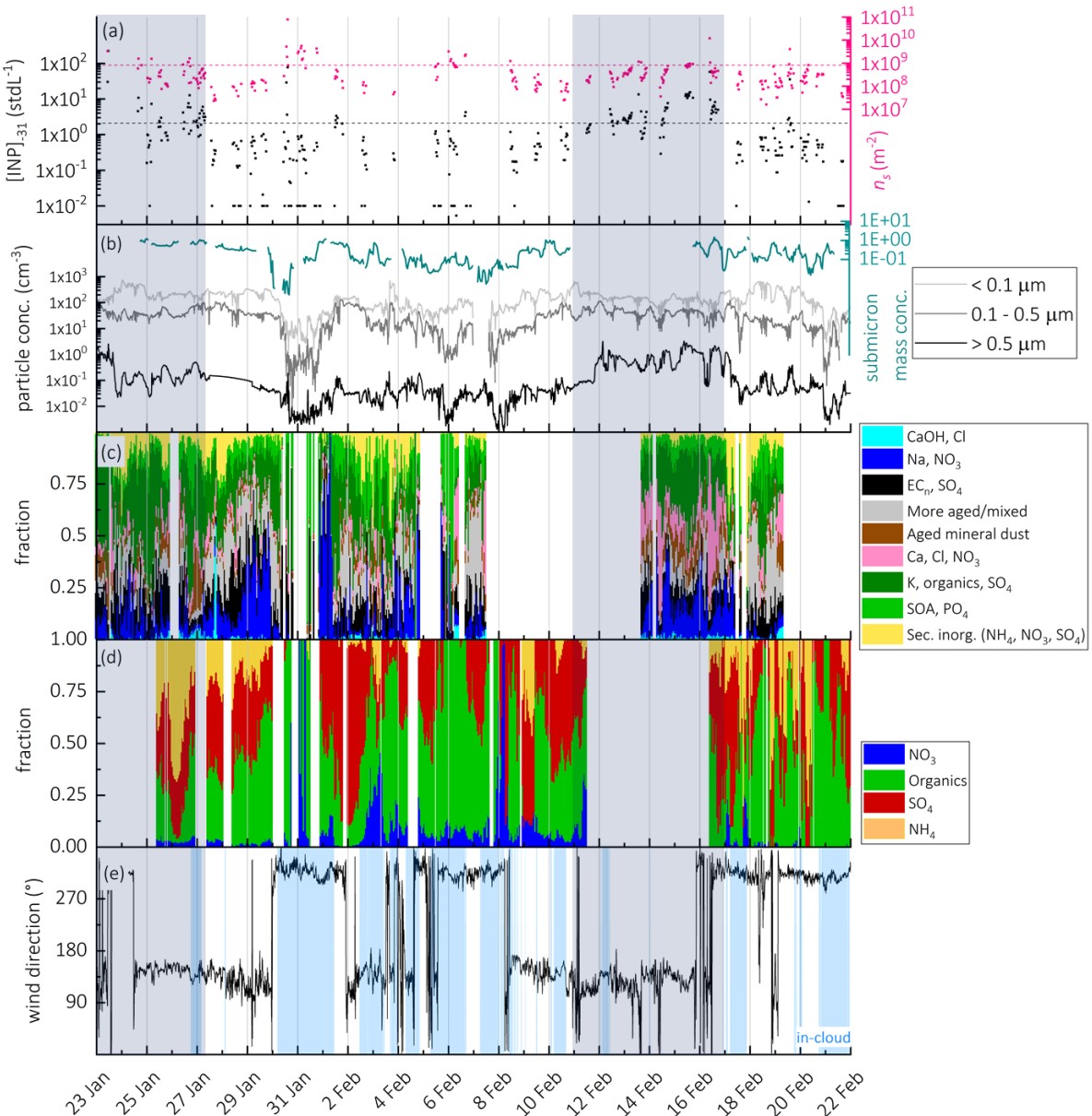

Figure 2: Timeseries of: panel a: [INP]$_{-31}$ (incl. INP<LOD) (black) and $n_s$ (based on INP>LOD) (pink) at -31 °C, determined with HINC (dashed lines indicate median [INP]$_{-31}$ and $n_s$ values from Lacher et al. (2018a)); panel b: particle number concentration for different size bins from the SMPS (< 0.5 µm) and OPC (> 0.5 µm), and submicron mass concentration derived from AMS measurements; panel c: relative abundance of particle types (based on particle number) during the campaign as measured with LAAPTOF; panel d: fractional contribution of the non-refractory aerosol compounds derived from the AMS measurements; panel e: wind direction and in-cloud conditions.

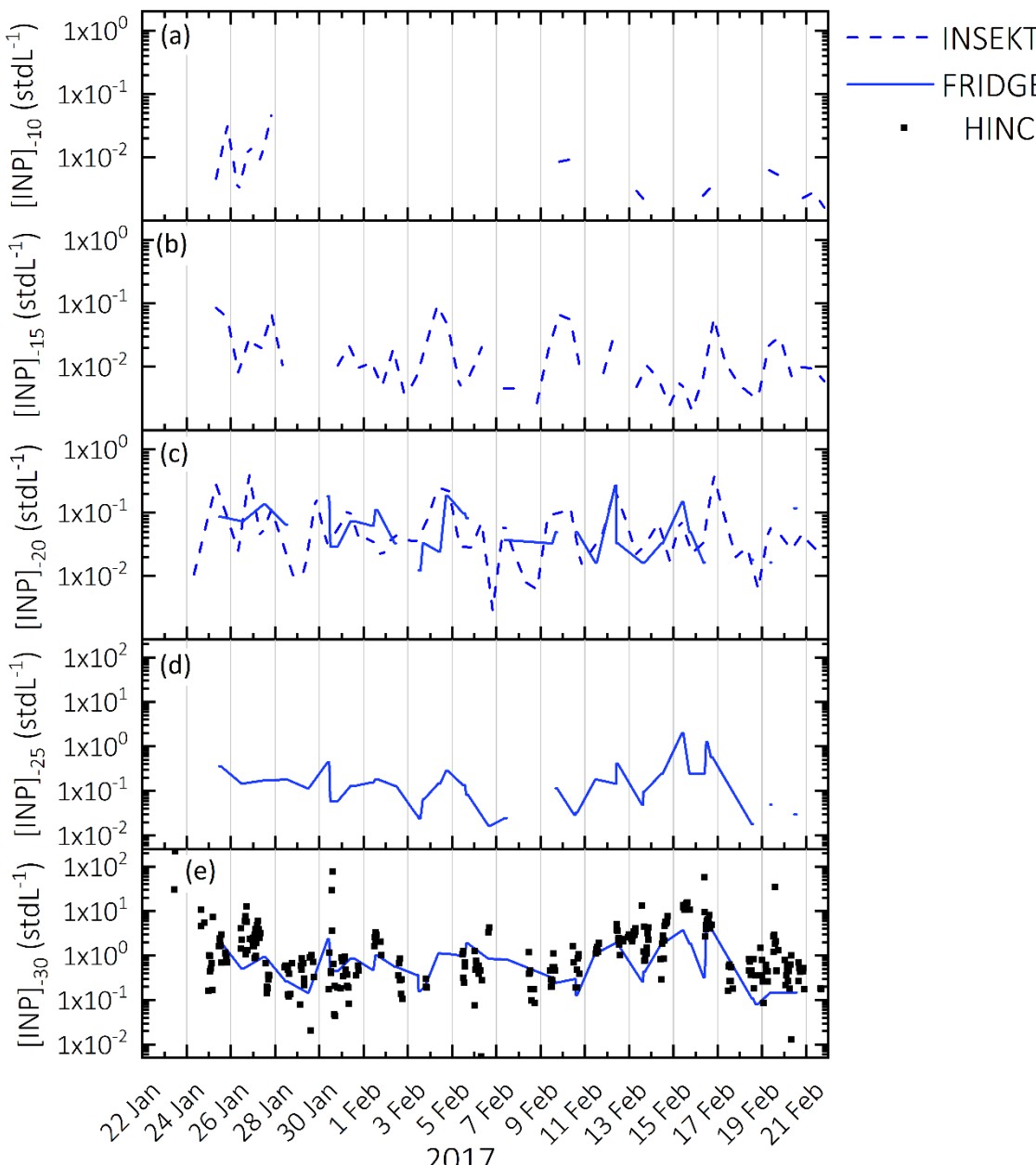

Figure 3: Timeseries of [INP] measured at (panel a) -10°C (INSEKT), (panel b) -15°C (INSEKT), (panel c) -20°C (INSEKT and FRIDGE), (panel d) -25°C (FRIDGE), and (panel e) -30/-31°C (FRIDGE/HINC respectively). Elevated [INP] during January 22 – 27 and February 11 – 17 are only detected at temperatures of -25°C and -30/-31°C.

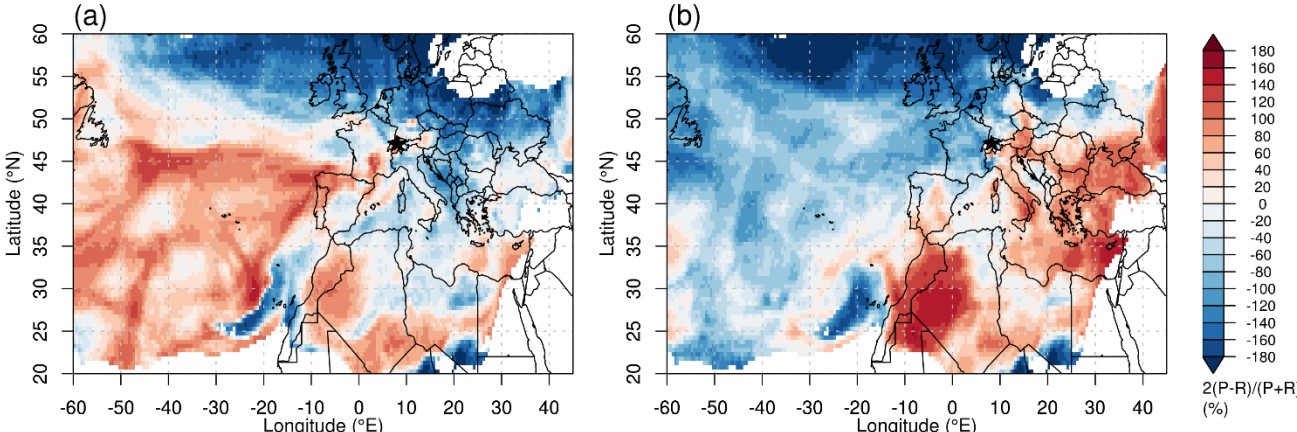

Figure 4: FLEXPART source emission sensitivities https://lagrange.empa.ch/FLEXPART_browser for particles released at JFJ (indicated by black star) every 3 hours and tracked backward in time, for time periods during this campaign with (panel a) elevated $n_s$ and (panel b) [INP], at -31° and above water saturation. Elevated values are above median values from a multi-year and multi-season analysis for free-tropospheric conditions at JFJ ($n_s > 8.19 \times 10^8$ m⁻² and [INP]₋₃₁ > 2.1 stdL⁻¹; Lacher et al., 2018a). The color code indicates the relative surface residence time, from low (blue) to high (red).

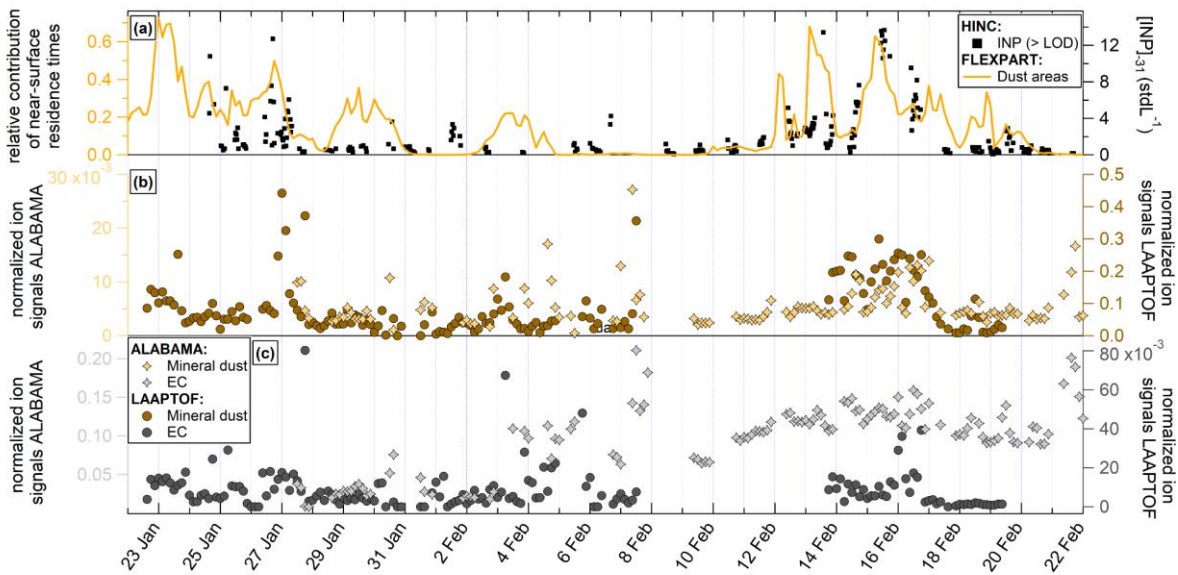

Figure 5: Time series of FLEXPART emission footprint sensitivities for desert dust surfaces (Northern Africa and Middle East, Sect. 2.7) along with INP concentration (panel a); timeline of single particle mass spectrometer marker ions for mineral dust particles (panel b), and EC marker ions (panel c) for both ALABAMA and LAAPTOF, respectively. SPMS reference spectra of the mineral dust and EC containing particle types are shown in Fig. S10 and S11.

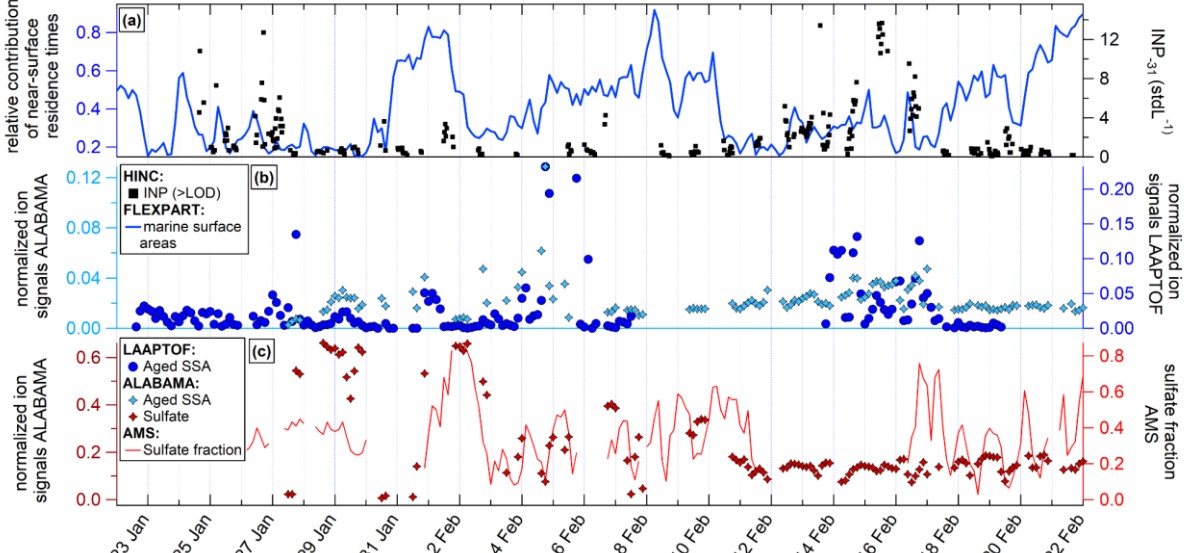

Figure 6: Time series of FLEXPART emission footprint sensitivities for marine surface areas along with INP concentration (panel a); timeline of mass spectrometer marker ions for aged sea spray aerosol (SSA) for both ALABAMA and LAAPTOF (panel b), and sulfate marker ions and sulfate fraction for ALABAMA and AMS (panel c). SPMS reference spectra of the aged sea spray and sulfate containing particle types are shown in Supplement (Fig. S9 and S10).

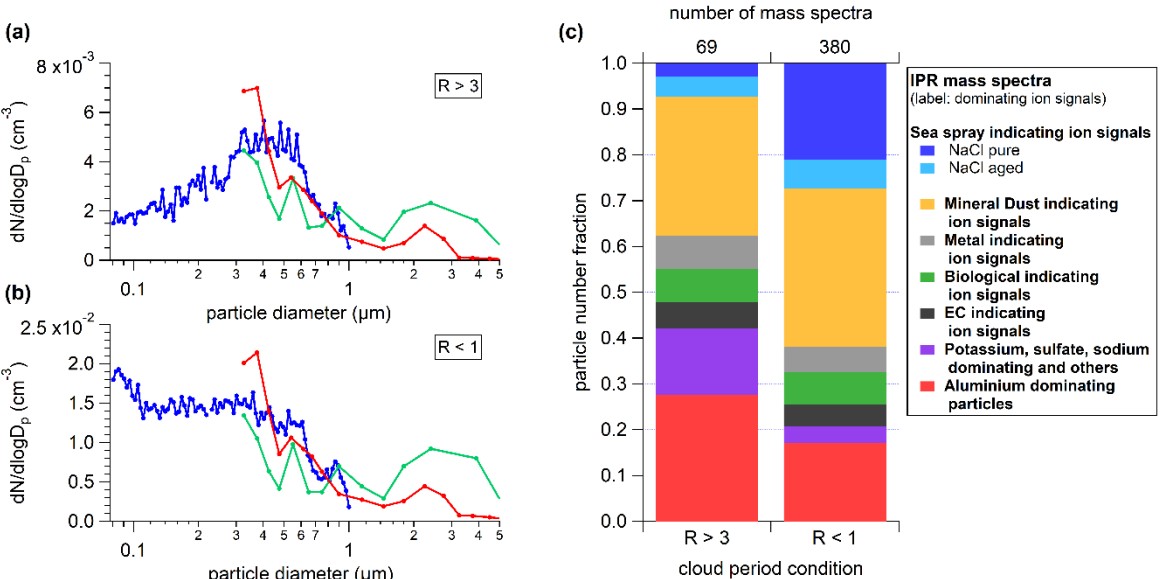

Figure 7: Comparison of duration weighted mean IPR number size distribution of cloud periods with lower ($R > 3$, panel a) and higher ($R < 1$, panel b) proportion of small IPR (<300 nm), and respective IPR composition (panel c), which is interpreted as lower and higher contribution of secondary ice, respectively (see description in section 3.5.1), Details about the individual cloud periods are presented in Tab.S3 and Tab.S4. The size distribution shown in panels a and b were measured using a UHSAS (blue), an OPS (green), and a Sky-OPC (red). The IPR composition shown in panel (c) was determined with the ALABAMA.

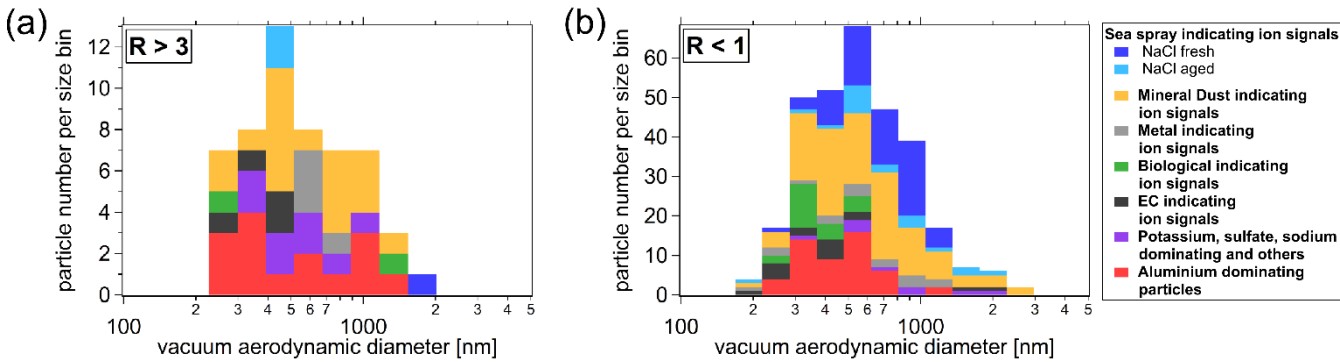

Figure 8: Size-resolved particle number fraction of the determined IPR types for cloud periods with lower ($R > 3$; panel a) and higher ($R < 1$; panel b) proportion of small IPR (<300 nm), which is interpreted as lower and higher contribution of secondary ice, respectively (see description in section 3.5.1).

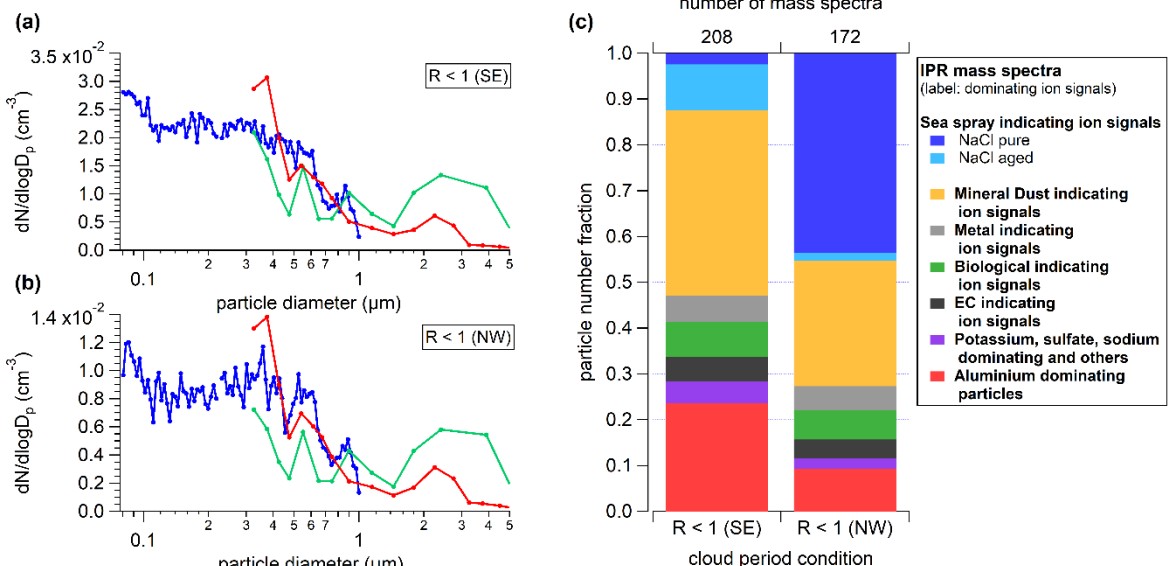

Figure 9: Comparison of the IPR number size distribution of cloud periods with higher ($R < 1$) proportion of small IPR (<300 nm), which is interpreted as higher contribution of secondary ice (see description in section 3.5.1) as a function of air mass flow direction, in the case of a south-easterly flow (SE, panel a) and north-westerly flow (NW, panel b), and respective IPR composition (panel c). The size distribution shown in panels and a and b were measured using a UHSAS (blue), an OPS (green), and a Sky-OPC (red). The IPR composition shown in panel c was determined with the ALABAMA.

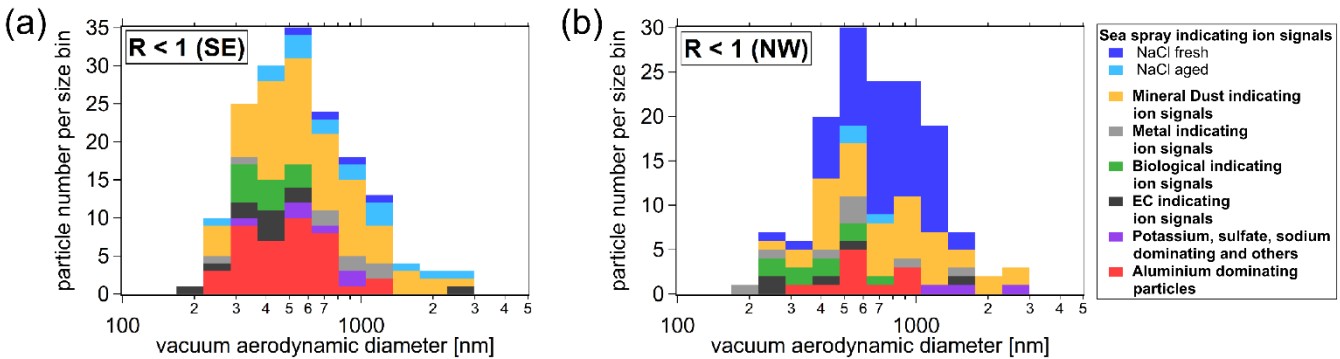

Figure 10: Size-resolved particle number fraction of the determined IPR types for cloud periods with higher proportion ($R < 1$) of small IPR (<300 nm), which is interpreted as a higher contribution of secondary ice, respectively (see description in section 3.5.1), selected by south-easterly flow (panel a) and north-westerly flow (panel b).

Table1: Ranked correlation coefficients for [INP]$_{-31}$ and respective $n_s$ (both determined > LOD) with m/z values for LAAPTOF (panel a) and ALABAMA (panel b), for method 1 and 2, respectively. The assignment of potential ions to m/z values can be found in the supporting information (Table S1).

(a) LAAPTOF

Method 1 cations / Method 1 anions

| | Method 1 cations r_pos | | | | Method 1 cations r_neg | | | | Method 1 anions r_pos | | | | Method 1 anions r_neg | | | |
|---|---|---|---|---|---|---|---|---|---|---|---|---|---|---|---|---|
| Rank | m/z | r([INP]$_{-31}$) | m/z | r($n_s$) | m/z | r([INP]$_{-31}$) | m/z | r($n_s$) | m/z | r([INP]$_{-31}$) | m/z | r($n_s$) | m/z | r([INP]$_{-31}$) | m/z | r($n_s$) |
| 1 | 46 | 0.51 | 56 | 0.36 | 29 | -0.42 | 13 | -0.30 | 60 | 0.59 | 60 | 0.39 | | | | |
| 2 | 45 | 0.50 | 46 | 0.34 | | | | | 76 | 0.55 | 76 | 0.36 | | | | |
| 3 | 44 | 0.47 | 96 | 0.34 | | | | | 77 | 0.53 | 35 | 0.35 | | | | |
| 4 | 57 | 0.47 | 74 | 0.32 | | | | | 37 | 0.51 | 77 | 0.35 | | | | |
| 5 | 56 | 0.46 | 57 | 0.31 | | | | | 35 | 0.49 | 72 | 0.32 | | | | |
| 6 | 74 | 0.44 | 45 | 0.30 | | | | | 59 | 0.47 | 37 | 0.32 | | | | |
| 7 | 96 | 0.43 | 44 | 0.30 | | | | | 120 | 0.46 | 120 | 0.32 | | | | |
| 8 | 58 | 0.42 | 41 | 0.29 | | | | | 50 | 0.45 | 50 | 0.32 | | | | |
| 9 | 75 | 0.42 | | | | | | | 43 | 0.44 | 16 | 0.31 | | | | |
| 10 | 76 | 0.41 | | | | | | | 28 | 0.44 | | | | | | |
| 11 | 81 | 0.37 | | | | | | | 16 | 0.44 | | | | | | |
| 12 | 82 | 0.36 | | | | | | | 17 | 0.43 | | | | | | |
| 13 | 83 | 0.35 | | | | | | | 72 | 0.43 | | | | | | |
| 14 | | | | | | | | | 42 | 0.43 | | | | | | |
| 15 | | | | | | | | | 36 | 0.41 | | | | | | |

Method 2 cations / Method 2 anions

| | Method 2 cations r_pos | | | | Method 2 cations r_neg | | | | Method 2 anions r_pos | | | | Method 2 anions r_neg | | | |
|---|---|---|---|---|---|---|---|---|---|---|---|---|---|---|---|---|
| Rank | m/z | r([INP]$_{-31}$) | m/z | r($n_s$) | m/z | r([INP]$_{-31}$) | m/z | r($n_s$) | m/z | r([INP]$_{-31}$) | m/z | r($n_s$) | m/z | r([INP]$_{-31}$) | m/z | r($n_s$) |
| 1 | 44 | 0.57 | 44 | 0.37 | 13 | -0.37 | 30 | -0.33 | 60 | 0.60 | 60 | 0.38 | | | | |
| 2 | 46 | 0.50 | 56 | 0.33 | | | | | 76 | 0.55 | 35 | 0.36 | | | | |
| 3 | 45 | 0.47 | 46 | 0.33 | | | | | 77 | 0.54 | 76 | 0.35 | | | | |
| 4 | 56 | 0.46 | 96 | 0.32 | | | | | 37 | 0.52 | 77 | 0.34 | | | | |
| 5 | 96 | 0.44 | 41 | 0.32 | | | | | 35 | 0.52 | 72 | 0.32 | | | | |
| 6 | 75 | 0.43 | 74 | 0.30 | | | | | 36 | 0.47 | 120 | 0.31 | | | | |
| 7 | 76 | 0.43 | | | | | | | 120 | 0.47 | 73 | 0.31 | | | | |
| 8 | 74 | 0.43 | | | | | | | 59 | 0.46 | 19 | 0.30 | | | | |
| 9 | 57 | 0.41 | | | | | | | 50 | 0.44 | | | | | | |
| 10 | 81 | 0.41 | | | | | | | 72 | 0.44 | | | | | | |
| 11 | 58 | 0.40 | | | | | | | 28 | 0.42 | | | | | | |
| 12 | 82 | 0.38 | | | | | | | 43 | 0.42 | | | | | | |
| 13 | 40 | 0.37 | | | | | | | 29 | 0.42 | | | | | | |
| 14 | 83 | 0.37 | | | | | | | 73 | 0.409 | | | | | | |

1275

(b) ALABAMA

### Method 1

|  | Method 1 cations | | | | | | | | Method 1 anions | | | | | | | |
|---|---|---|---|---|---|---|---|---|---|---|---|---|---|---|---|---|
|  | r_pos | | | | r_neg | | | | r_pos | | | | r_neg | | | |
| Rank | m/z | r([INP]$_{-31}$) | m/z | r($n_s$) | m/z | r([INP]$_{-31}$) | m/z | r($n_s$) | m/z | r([INP]$_{-31}$) | m/z | r($n_s$) | m/z | r([INP]$_{-31}$) | m/z | r($n_s$) |
| 1 | 165 | 0.47 | 78 | 0.44 | | | | | 12 | 0.48 | 42 | 0.47 | 141 | -0.53 | 97 | -0.39 |
| 2 | 78 | 0.46 | 144 | 0.42 | | | | | 18 | 0.45 | 43 | 0.43 | 99 | -0.51 | 195 | -0.37 |
| 3 | 7 | 0.44 | 165 | 0.41 | | | | | 37 | 0.44 | 12 | 0.42 | 97 | -0.49 | | |
| 4 | 35 | 0.43 | 62 | 0.39 | | | | | 35 | 0.44 | 27 | 0.40 | 155 | -0.46 | | |
| 5 | 62 | 0.41 | 63 | 0.37 | | | | | 4 | 0.44 | 37 | 0.39 | 96 | -0.45 | | |
| 6 | 22 | 0.40 | 222 | 0.37 | | | | | 9 | 0.43 | 35 | 0.37 | 217 | -0.42 | | |
| 7 | 23 | 0.39 | 224 | 0.37 | | | | | 23 | 0.43 | 9 | 0.37 | 115 | -0.40 | | |
| 8 | 133 | 0.38 | 132 | 0.36 | | | | | 34 | 0.43 | 40 | 0.34 | 101 | -0.40 | | |
| 9 | 164 | 0.38 | 23 | 0.36 | | | | | 50 | 0.39 | | | | | | |
| 10 | 144 | 0.37 | 151 | 0.36 | | | | | 3 | 0.39 | | | | | | |
| 11 | 224 | 0.35 | 108 | 0.35 | | | | | | | | | | | | |
| 12 | 132 | 0.35 | 202 | 0.35 | | | | | | | | | | | | |
| 13 | 222 | 0.34 | 217 | 0.35 | | | | | | | | | | | | |
| 14 | 118 | 0.34 | 240 | 0.34 | | | | | | | | | | | | |
| 15 | 20 | 0.34 | 181 | 0.34 | | | | | | | | | | | | |
| 16 | 151 | 0.34 | 118 | 0.34 | | | | | | | | | | | | |
| 17 | | | 180 | 0.34 | | | | | | | | | | | | |
| 18 | | | 223 | 0.33 | | | | | | | | | | | | |
| 19 | | | 231 | 0.33 | | | | | | | | | | | | |
| 20 | | | 213 | 0.33 | | | | | | | | | | | | |
| 21 | | | 133 | 0.32 | | | | | | | | | | | | |

### Method 2

|  | Method 2 cations | | | | | | | | Method 2 anions | | | | | | | |
|---|---|---|---|---|---|---|---|---|---|---|---|---|---|---|---|---|
|  | r_pos | | | | r_neg | | | | r_pos | | | | r_neg | | | |
| Rank | m/z | r([INP]$_{-31}$) | m/z | r($n_s$) | m/z | r([INP]$_{-31}$) | m/z | r($n_s$) | m/z | r([INP]$_{-31}$) | m/z | r($n_s$) | m/z | r([INP]$_{-31}$) | m/z | r($n_s$) |
| 1 | 60 | 0.61 | 60 | 0.55 | | | | | 76 | 0.54 | 42 | 0.40 | 155 | -0.51 | 97 | -0.36 |
| 2 | 51 | 0.61 | 132 | 0.52 | | | | | 12 | 0.50 | 76 | 0.39 | 97 | -0.51 | 177 | -0.35 |
| 3 | 24 | 0.61 | 39 | 0.47 | | | | | 35 | 0.49 | 12 | 0.38 | 141 | -0.50 | 195 | -0.34 |
| 4 | 12 | 0.57 | 78 | 0.44 | | | | | 37 | 0.46 | 9 | 0.37 | 106 | -0.44 | | |
| 5 | 62 | 0.57 | 63 | 0.44 | | | | | 19 | 0.45 | 43 | 0.37 | 99 | -0.44 | | |
| 6 | 36 | 0.57 | 84 | 0.44 | | | | | 32 | 0.44 | 35 | 0.34 | 233 | -0.41 | | |
| 7 | 23 | 0.55 | 108 | 0.43 | | | | | 18 | 0.44 | 78 | 0.34 | | | | |
| 8 | 40 | 0.55 | 62 | 0.43 | | | | | 4 | 0.44 | 37 | 0.33 | | | | |
| 9 | 78 | 0.54 | 36 | 0.42 | | | | | 77 | 0.44 | 19 | 0.33 | | | | |
| 10 | 165 | 0.54 | 120 | 0.41 | | | | | 9 | 0.43 | | | | | | |
| 11 | 84 | 0.53 | 240 | 0.39 | | | | | | | | | | | | |
| 12 | 63 | 0.52 | 23 | 0.38 | | | | | | | | | | | | |
| 13 | 132 | 0.52 | 91 | 0.38 | | | | | | | | | | | | |
| 14 | 35 | 0.50 | 165 | 0.38 | | | | | | | | | | | | |
| 15 | 67 | 0.49 | 24 | 0.38 | | | | | | | | | | | | |
| 16 | 22 | 0.46 | 76 | 0.38 | | | | | | | | | | | | |
| 17 | 133 | 0.45 | 41 | 0.37 | | | | | | | | | | | | |
| 18 | 17 | 0.43 | 138 | 0.37 | | | | | | | | | | | | |
| 19 | 120 | 0.43 | | | | | | | | | | | | | | |