# Peer review of "Sources and nature of ice-nucleating particles in the free troposphere at Jungfraujoch in winter 2017"

_Atmospheric Chemistry and Physics, 2021_

## Referee Comment (RC2)

**Anonymous Review of *Sources and nature of ice-nucleating particles in the free troposphere at Jungfraujoch in winter 2017**

Anonymous Reviewer

July 2021

**1    Summary**

In this work, Lacher et al., measured ice nucleating particle (INP) concentrations at -31 °C at the high-altitude research station atop Jungfraujoch (JFJ). INP concentrations were measured semi-continuously for one month (Jan. 22 - Feb 22., 2017) using the Horizontal Ice Nucleation Chamber (HINC). To increase large particle concentrations in the free troposphere, a Portable Fine Particle Concentrator was used. Finally, offline INP concentration measurements were taken to access warmer temperatures in the INP spectra. In addition to INP concentrations, the authors also measured aerosol size distributions, single-particle composition (LAAPTOF, ALABAMA), and bulk aerosol composition (AMS) in parallel. In addition to these aerosol measurements, size distributions and single-particle composition (ALABAMA) were measured behind an ice-selective counterflow virtual impactor (ice-CVI). Finally, back trajectory models were used to help define the sources of INP.

Overall, this work contains an impressive amount of information, which may be helpful to elucidate the role of aerosol size and composition on atmospheric ice nucleation. The HINC instrument is well-known to the ice nucleation community, and has published several "long-term measurement "publications." Much of the measurements are taken in the free troposphere, which is more relevant to cold-cloud formation that most ground-site operations. The paper itself is well-written, and only has a few technical corrections.

Most of the analysis is interesting, I particularly enjoyed the FLEXPART + single-particle mass spec. analyses, as well as the IPR measurements; however, several of the analyses may need further investigation. This are outlined in the general comment section. Most of the general comments are about the Spearman's rho analyses in Figure 5.

**2    General Comments**

- The comparison between these methods is not quite apples-to-apples. The size ranges and detection limits of all of these instruments are quite different. The authors do a good job of describing this problem in Section 2.3.1, but I feel the authors should add a Figure of "Detection Efficiency vs. Size," and add curves for at least HINC (with and without the PFPC), LAAPTOF, and ALABAMA.

- It is really hard to determine what the Spearman's rho values actually mean. While it's a well-known equation, it's main purpose is to detect if the relationship between two variables is monotonic. In this paper, it is used to define a correlation between INP concentrations and other aerosol measurements taken in parallel. It's a subtle distinction, and likely matters little if the rho values are very high (say $> 0.8$), but it becomes difficult to envision what a rho of 0.5 really means in this context. I suggest that the authors spend some time defining the equation and also plotting some rank correlations in the supplemental to help the reader envision how good / poor these rank correlatons are.

- Likewise, I'm not sure that taking the mean $\pm$ std. dev. of the rank correlations really defines what m/z have "significant" correlations. It makes the assumption that, in general, peaks in single-particle mass spec. are not correlated with INP concentrations. This assumption was not rigorously tested, at least in this paper, and I see no good reason why it would be true. For example, Mineral Dust spectra contain many relatively unique peaks; ostensibly, these peaks will correlate with INP concentrations.

- One of the most troubling aspects of this paper is that some of its results contradict themselves. For example, Figure 5 indicates that sea-spray aerosol (SSA) are correlated with INP; however, Figure 7 indicates that the opposite is true. One reason for this is that many peaks are not unique to a single particle type. Thus, trying to attribute a peak to single particle type as done in Figure 5 may give erroneous results. For example, m/z 60 is defined as mineral dust ($SiO_2^+$ for LAAPTOF, but elemental carbon ($C_5^+$) for ALABAMA. I suspect that it shows up in each particle type for both single-particle mass specs.

**3    Minor Comments**

- Some Figures are extremely hard to read in print format. I can not read the axes in Figure 4, some text and the "markers" in part C of Figure 5, and the legends in Figures 9 and 11.

- Line 155: A detailed description of how the concentration factor needs to be added here. To say that a you increase aerosol ¿100 by a factor of 20 is greatly simplifying what is happening.

- Line 160: The caveats of converting ambient [INP] to $n_s$ should be addressed here. By definition, $n_s$ assumes that ice nucleation is deterministic, but it has been shown that this is not necessarily true for aerosol populations with a wide variety of ice-active site densities.

- Line 245: You are biased by the mass spec. detection efficiencies. Was there any attempt at normalization to the optical particle counters?

- Line 469: This statement is incorrect for two reasons. One–a large signal in one particle could greatly skew the average because the signal spans several orders of magnitude and a linear average was applied. That is one INP disproportionately affecting the signal average. Two–a single peak in a mass spectrum may qualitatively scale with abundance of that element or fragment, but it likely does not scale with the abundance of a "substance" or ice-active site.

- Line 537: I believe that dust particles fluoresce slightly in the WIBS. This is one reason why the FBAP thresholds are so strict. This should be mentioned here–or it should be mentioned why dust may not be suspected.

- Line 626: I'm not quite sure how you arrive at the conclusion that 70% of EC particles also contain mineral dust. From general aerosol knowledge, this seems like a vast overestimation. Thus, more details are need to support this statement.

**4 Technical Comments**

- Line 42: Are metallic particles and biological particles each 10% or are they 10% together?

- Line 60: I would not say that INP concentrations are entirely "unconstrained." Plenty of papers show a reasonable range of ambient INP concentrations depending on the air mass.

- Line 116: Is there an estimate of the transport time from the MBL to JFJ?

- Line 141: I think it should be "[INP] are naturally low ..."

- Line 141: This sentence could be split into two sentences. The phrase "but is generally higher ..." it difficult to follow because there are many potential subjects –e.g., nucleation temperature, detection limit, [INP], etc.

- Line 190: Should it be "into" instead of "onto?"

- Line 200: Should it be "decreases" instead of "decreased?"

- Line 206: Should it be "mJ / pulse?"

- Line 289: What is the typical size range of ice crystals at JFJ?

- Line 343: There is a lot of information in Figure 2, so it would be instructive to highlight these periods in Figure 2.

- Line 353: This reference to the Kammermann paper is not really supported by Figure 9.

- Line 483: Please move the legends in Figures 6 and 7, they are obstructing one of the high INP periods.

- Line 563: More secondary ice from your calculations, right? The way this sentence reads, it sounds like it's not an inferred measurement.

---

## Author Comment (AC1)

**Author comment in response to the comments provided by Referee #1**

We thank Referee #1 for their effort in reading and commenting our manuscript and giving helpful feedback. In the following, we repeat the **referee's comments in bold typeface**, and give point-by-point answers in normal typeface; extracts from the *original manuscript are presented in red italic*, and from the *revised manuscript in blue italic*. Line numbers are referring to the updated manuscript version.

General Comments:

**Due to the low and large variability of the concentrations in the atmosphere, our understanding of the chemical composition and sources of ice nucleating particles is still quite limited. This article investigated the potential chemical characteristics and sources of ice nucleating particles (INPs) at the high-altitude research station Jungfraujoch (JFJ) in winter 2017 based on measurements from a suite of sophisticated instruments and the FLEXPART source emission model calculations. Their results show that mineral dust and particles of marine origin are the most important contributors to ice nucleation. These are supported by analysis of airmass back trajectories and the chemical composition of ice particle residuals. The results are very important and add some new insights into the understanding of possible chemical composition and sources of INPs. The article is generally well written, concise and should be publishable if the following specific comments and suggestions can be considered in revision.**

We appreciate these general remarks regarding our manuscript and believe that we have addressed the comments and suggestions provided by Referee #1 in the revised manuscript as indicated below.

**The authors used the INPs concentration at -31 °C to represent ice formation in mixed-phase clouds. I think this temperature is too low for typical mixed-phase clouds. Actually, the authors have also indicated in Line 367, that temperature ranged from -5 °C to -18 °C to be a relevant range in which MPCs can form.**

The reviewer addresses two points here. First, mixed-phase clouds can form within the temperature range from 0 °C to approximately -38 °C (e.g., Korolev et al., 2017). Thus, our chosen nucleation temperature of -31 °C albeit toward the lower end, is still very relevant for mixed-phase cloud formation, but perhaps for a higher altitude than where we measured at the JFJ.

Second, indeed, the mixed-phase clouds present at Jungfraujoch are warmer than -31 °C. Field INP concentration measurements are typically performed at temperatures colder than the ambient conditions, because they investigate the *potential* of the aerosol population to nucleate ice if the air mass containing those particles would experience this temperature and supersaturated conditions. The great advantage of our sampling location and sampling time in the winter months is that the measurements were taken in the free troposphere, which is not impacted by local emissions but of long-range air masses, and thus is better representative of

aerosol particles of a global burden (Lacher et al., 2018a). Thus we believe that our INP concentration measurements at -31 °C are relevant for cloud microphysical properties in the free troposphere and not only locally at the JFJ. When measuring at the ambient temperature or warmer, it is further possible that the INPs are depleted due to a pre-activation during the transport from INP source regions to the measurement site (Conen et al., 2015). At a measurement site as Jungfraujoch, which is far away from INP source regions, it is especially likely that this process takes place.

Last but not least, the ambient INP concentrations at Jungfraujoch are typically very low, and by choosing a nucleation temperature as low as -31 °C, we ensure that most of our measurements are still above the detection limit of HINC (Lacher et al., 2017).

We clarify this aspect by mentioning in lines 140 - 141:

*The $RH_w$ of 103% ensures that the entire aerosol layer which experiences a varying RH between $101 - 103\%$ is above water saturation such that the particles can activate into droplets in the given residence time of HINC.*

**A reference by Jiang et al. (2016) should be added in Line 70, it provided another evidence that the INP concentration increased from about 10 per liter to more than a hundred per liter.**

Thank you for this suggestion. We include now the study by Jiang et al. (2016) to reference the importance of mineral dust for ice nucleation (see lines 63 - 64 in revised manuscript version).

**In Line 95, after Murphy et al., I suggest to add Chen et al. (2021), who used a single particle mass spectrometer together with a wide-range aerosol particle spectrometer to determine the possible chemical components and sources of INPs.**

Thank you for pointing at this interesting study. We include Chen et al. (2021) now in the statement in line 85 of the revised manuscript.

**Line 149: "-31 °C (±0.4 °C; [INP]-31) and at a relative humidity with respect to water of 103% (±2%), representing condensation/immersion freezing, relevant for the formation of MPCs": As has pointed out above, -31°C might be lower than typical temperatures in mixed-phase clouds, and RH of 103% is possibly much higher than general winter orographic clouds.**

Please see our comment above regarding the temperature range for mixed-phase clouds. Indeed, a relative humidity with respect to water ($RH_w$) of 103% is higher than ambient supersaturation conditions in mixed-phase clouds. There are three reasons for this. First, HINC has an uncertainty in $RH_w$ of 2%, and by setting a $RH_w$ of 103% we ensure that the entire aerosol layer in the chamber is exposed to conditions of or above water saturation. Some of this uncertainty arises from the fact that the part of the aerosol layer closer to the warm wall will experience $RH_w$ below 103%. By maintaining the nominal $RH_w$ at 103%, we ensure that even this part of the aerosol layer is still at or above $RH_w = 100\%$. Second, the residence time of particles in continuous flow diffusion chambers is on the order of a few seconds, which limits

the activation and growth time for cloud droplets and ice crystals; by creating such a high supersaturation, we are counteracting this time effect. Moreover, a high supersaturation also favors the activation of all CCN (and INPs) into cloud droplets, even in the case of hydrophobic particles and high CCN number concentrations (see also the discussion in Rogers et al. (2001)).

**Lines 174-175: "which requires the surface area distribution concentration to be calculated from the number size distribution.": How the surface areas of aerosol particles were calculated? The assumption of aerosol particle shape could significantly impact on the calculated surface area.**

The reviewer is right that the aerosol particle shape can impact the calculated aerosol particle surface area. For our calculation we assumed a spherical particle shape, as the true particle shape was not measured. We add a statement that our assumption might lead to increased uncertainties in lines 164 - 167:

*Measurements from the GAW SMPS and OPS are used to calculate the available surface area of the aerosol population by assuming a spherical particle shape and that the refractive index of the ambient aerosol population is represented by the calibrated value of the OPS. We acknowledge that this can lead to higher uncertainties, which are not quantified here.*

**Lines 211-212: "The overall detection efficiency of the LAAPTOF is between 0.01 (±0.01) % to 4.2 (±2.4) %": Is it so low?**

Yes, that is correct. We acknowledge that due to this low detection efficiency in LAAPTOF the large majority of the particles is not analyzed; however, we still consider that the measured particles are representative for the characteristics of the overall aerosol properties and for their change over time, which is used in this study to investigate parallel changes in the INP population.

During the review process we realized that the values stated from ALABAMA and LAAPTOF were not referring to the same detection efficiencies; while the values given for LAAPTOF are the *overall detection efficiencies*, which we define as detection efficiency combined with the hit rate, the ALABAMA stated only the detection efficiency. Moreover, we added a graph on the size-dependent detection efficiencies from the two single particle mass spectrometers and HINC, with and without the PFPC. As our aim is to investigate size dependence of detection efficiency in relative terms, we normalized those measurements to the maximum value in each instrument.

We updated the manuscript accordingly in lines 207 - 225, and include Fig. S1 in the appendix:

*The overall detection efficiency (combining the detection efficiency and the hit rate) of the LAAPTOF is between 0.01 (±0.01) % to 4.2 (±2.4) %, in the size range of 0.2 to 2 µm based on polystyrene latex particles (PSL). The highest overall detection efficiency is for 1µm and lowest for 2 µm (Shen et al., 2018). Note that such efficiency is also particle type dependent (Shen et al., 2018, 2019). More details on the LAAPTOF can be found in Gemayel et al. (2016), Reitz et al. (2016), and Shen et al. (2018, 2019). Details on the ALABAMA have been presented in Brands et al. (2011), Roth et al. (2016), Schmidt et al. (2017), and Clemen et al. (2020). The detection efficiency of the ALABAMA during this campaign was between 40% and*

*60% in the size range of 0.3 to 1.0 μm based on PSL particles. Up to a particle size of about 1.3 μm, the detection efficiency of the ALABAMA decreased to less than 30% and is estimated to be about 5 (±5) % for 2 μm. At the same time, the hit rate during those tests using PSL particles was lower, such that the overall detection efficiency for the ALABAMA was only between 1% and 16% in the size range from 0.3 to 1 μm. As those values are based on measurements using PSL particles, they can vary considerably during field applications; e.g., the ALABAMA hit rates were significantly higher than those of the PSL test measurements (which is attributed to particle charge effects during the nebulization of the PSL particles). In the light of our research objectives, focusing on the general trend of the aerosol particle composition, we therefore provide an overview of the size dependent overall detection efficiency from the LAAPTOF and the ALABAMA normalized to the maximum value measured, together with the normalized transmission efficiency from HINC (Fig. S1). From those normalized values it is visible that both SPMSs detect aerosol particles in the same size range, with a maximum between 0.5 and 1 μm, and therefore yield comparable information on the particle composition in this size range. HINC measures particles below 2 μm with a high efficiency which can have an impact on the comparison between the INP measurements from HINC and the results obtained from the ALABAMA and the LAAPTOF.*

[Figure]

*Figure S1: Normalized size-dependent detection efficiencies for aerosol particles in ALABAMA and LAAPTOF), and normalized transmission efficiency for HINC, and for HINC at the PFPC (based on calculations using the measured transmission efficiency with size-dependent enrichment from the PFPC (Gute et al., 2019)); the measurements were normalized to the maximum detection efficiencies of 4.3% (LAAPTOF), 15.6% (ALABAMA), 100% (HINC), and 1480% (HINC at PFPC).*

**Line 366-367: "The ambient temperature ranged from -5 °C to -18 °C (Fig. S3, panel b), which is a relevant range in which MPCs can form": If the ambient temperature ranged from -5°C to -18°C, why choose -31°C as the nucleation temperature for INP analysis?**

Please see our answer to the first comment. In addition, we would expect that any aerosol particles active as INP in the range -5 to -18 °C will also be active at -31 °C thus we should capture these particles in our reported INP concentrations.

**Line 396-397: "particle types "secondary inorganics", "K, organic sulfate", and "more mixed/aged" measured by the LAAPTOF…": "secondary inorganics" and "organic sulfate" were not shown in the figure. Please keep consistent with the chemicals shown in the figure.**

Secondary inorganics are referring to $NH_4$, $NO_3$, $SO_4$; we include "secondary inorganics" now in the legend of Fig. 2, panel c. Indeed, it should read "K, organics, sulfate", thank you for pointing this out. We updated the sentence in line 391:

*...particle types "secondary inorganics ($NH_4$, $NO_3$, $SO_4$)", „K, organics, sulfate"...*

Please note that Fig. 2 uses particle type clusters based on fuzzy c-means, in order to give a general overview of the time series of particle composition during the campaign. Figure 5, on the other hand, assigns the best correlating marker ions to a particle type.

**Lines 477-479: "Such compounds are typically water soluble and are therefore not expected to contribute to immersion freezing INPs at conditions assessed in HINC": Negative correlations means the present of those chemicals would inhibit ice formation.**

Indeed, this finding can be interpreted that those chemicals inhibit ice formation, which is, however, unlikely as any potential coating does not impact the ice nucleation ability in the immersion freezing mode (e.g., Kulkarni et al., 2014; Kanji et al., 2019) but can inhibit ice nucleation in the deposition or pore condensation freezing mode. Thus, we did not interpret this finding to be an effect of ice nucleation inhibition, but of an air mass containing a higher fraction of such water soluble aerosol particles which are all together not ice-active. We add this discussion to the manuscript in lines 471 - 472:

*Moreover, we do not expect that those components reduced the ice nucleation ability of the ambient aerosols by a coating effect in the immersion freezing regime (e.g., Kulkarni et al., 2014; Kanji et al., 2019).*

**Lines 576-578: "Thus, the timeseries of the ALABAMA and AMS sulfate measurements are anticorrelated to the [INP]-31 time series which is not surprising given soluble aerosol particles are not expected to contribute to heterogeneous ice nucleation above -38 °C": As has been pointed out above, negative correlation means that the present of sulfate would suppress those particles to act as INPs.**

Please see our answer to the comment above.

**Line 637: mineral dust particles can also be incorporated into cloud particles when coated dust particles serves as CCN or collected by cloud particles, not necessarily as INPs, especially for cloud formed at warmer temperatures (>-17°C as indicated in the text).**

We agree that the IPR analysis can be impacted by mineral dust particles when sampling secondary formed ice crystals, which originally were cloud droplets and might contain mineral dust. However, we see a strong mineral dust particle signal sampling different cloud events, and do not observe this highly variable fraction as we did for sea spray indicating ions.

**Line 739: remove "immersion freezing", since some of the instruments, such as FRIDGE, only measure INPs of deposition or condensation-freezing mode.**

Thank you for this attentive correction, agreed and changed.

Technical corrections

**Line 55: remove "the" after "by";**

We have modified the statement to read (lines 48 - 49) "*... they can modulate the microphysical properties of cirrus and mixed-phase clouds (MPCs)* by initiating *ice crystal formation.*"

**Line 81 and Line 103: change "as e.g." to "such as that";**

Agreed and changed.

**References, such as Eriksen Hammer et al. (2018) in Line 113, Collaud Coen et al. (2004) in Line 125 and 127, 2013; Pandey Deolal et al. (2014) in Line 330, should be cited with the last name of the first author;**

These are the last names of the first authors, please see the respective publication details.

**Line 189: remove "for analysis";**

Corrected.

**Line 383: remove "then";**

We removed "then".

**Line 387: Should "FRIDGE and INSEKT" be "FRIDGE and HINC"?**

Thank you, this is correct, we changed the statement accordingly.

**Line 639: add "by" after "caused".**

Agreed and changed.

References

Conen, F., Rodríguez, S., Hüglin, C., Henne, S., Herrmann, E., Bukowiecki, N., and Alewell, C.: Atmospheric ice nuclei at the high-altitude observatory Jungfraujoch, Switzerland, Tellus B, 67, 10.3402/tellusb.v67.25014, 2015.

Lacher, L., Lohmann, U., Boose, Y., Zipori, A., Herrmann, E., Bukowiecki, N., Steinbacher, M., and Kanji, Z. A.: The Horizontal Ice Nucleation Chamber (HINC): INP measurements at conditions relevant for mixed-phase clouds at the High Altitude Research Station Jungfraujoch, Atmos. Chem. Phys., 17, 15199-15224, 10.5194/acp-17-15199-2017, 2017.

Lacher, L., DeMott, P. J., Levin, E. J. T., Suski, K. J., Boose, Y., Zipori, A., Herrmann, E., Bukowiecki, N., Steinbacher, M., Gute, E., Abbatt, J. P. D., Lohmann, U., and Kanji, Z. A.: Background Free-Tropospheric Ice Nucleating Particle Concentrations at Mixed-Phase Cloud Conditions, J. Geophys. Res. Atmos. , 123, 10,506-510,525, 10.1029/2018JD028338, 2018a.

Korolev, A., McFarquhar, G., Field, P. R., Franklin, C., Lawson, P., Wang, Z., Williams, E., Abel, S. J., Axisa, D., Borrmann, S., Crosier, J., Fugal, J., Krämer, M., Lohmann, U., Schlenczek, O., Schnaiter, M., and Wendisch, M.: Mixed-Phase Clouds: Progress and Challenges, Meteorol. Monogr., 58, 5.1-5.50, 10.1175/amsmonographs-d-17-0001.1, 2017.

Kanji, Z. A., Sullivan, R. C., Niemand, M., DeMott, P. J., Prenni, A. J., Chou, C., Saathoff, H., and Möhler, O.: Heterogeneous ice nucleation properties of natural desert dust particles coated with a surrogate of secondary organic aerosol, Atmos. Chem. Phys., 19, 5091-5110, 10.5194/acp-19-5091-2019, 2019.

Kulkarni, G., Sanders, C., Zhang, K., Liu, X., and Zhao, C.: Ice nucleation of bare and sulfuric acid-coated mineral dust particles and implication for cloud properties, Journal of Geophysical Research: Atmospheres, 119, 9993-10011, doi.org/10.1002/2014JD021567, 2014.

Rogers, D. C., DeMott, P. J., Kreidenweis, S. M., and Chen, Y.: A Continuous-Flow Diffusion Chamber for Airborne Measurements of Ice Nuclei, J. Atmos. Oceanic Technol., 18, 725-741, 10.1175/1520-0426(2001)018<0725:ACFDCF>2.0.CO;2, 2001.

---

## Author Comment (AC2)

**Author comment in response to the comments provided by Referee #2**

We thank Referee #2 for commenting our manuscript and giving valuable suggestions. In the following, we repeat the **referee's comments in bold typeface**, and give point-by-point answers in normal typeface; extracts from the *original manuscript are presented in red italic*, and from the *revised manuscript in blue italic*. Line numbers are referring to the updated manuscript version.

**In this work, Lacher et al., measured ice nucleating particle (INP) concentrations at -31 °C at the high-altitude research station atop Jungfraujoch (JFJ). INP concentrations were measured semi-continuously for one month (Jan. 22 – Feb 22., 2017) using the Horizontal Ice Nucleation Chamber (HINC). To increase large particle concentrations in the free troposphere, a Portable Fine Particle Concentrator was used. Finally, offine INP concentration measurements were taken to access warmer temperatures in the INP spectra. In addition to INP concentrations, the authors also measured aerosol size distributions, single-particle composition (LAAPTOF, ALABAMA), and bulk aerosol composition (AMS) in parallel. In addition to these aerosol measurements, size distributions and single-particle composition (ALABAMA) were measured behind an ice-selective counterflow virtual impactor (ice-CVI). Finally, back trajectory models were used to help define the sources of INP.**

**Overall, this work contains an impressive amount of information, which may be helpful to elucidate the role of aerosol size and composition on atmospheric ice nucleation. The HINC instrument is well-known to the ice nucleation community, and has published several "long-term measurement "publications." Much of the measurements are taken in the free troposphere, which is more relevant to cold-cloud formation that most ground-site operations. The paper itself is well-written, and only has a few technical corrections.**

**Most of the analysis is interesting, I particularly enjoyed the FLEXPART + single-particle mass spec. analyses, as well as the IPR measurements; however, several of the analyses may need further investigation. This are outlined in the general comment section. Most of the general comments are about the Spearman's rho analyses in Figure 5.**

Thank you for these remarks about our manuscript. We improved the presentation, discussion, and interpretation of our analysis according to your comments and suggestions.

General comments

**The comparison between these methods is not quite apples-to-apples. The size ranges and detection limits of all of these instruments are quite different. The authors do a good job of describing this problem in Section 2.3.1, but I feel the authors should add a Figure**

**of "Detection Efficiency vs. Size," and add curves for at least HINC (with and without the PFPC), LAAPTOF, and ALABAMA.**

We agree with this suggestion and have done that in Fig S1. In addition, during the review process we realized that the values stated from the ALABAMA and the LAAPTOF were not referring to the same detection efficiencies; while the values given for LAAPTOF are *the overall detection efficiencies*, which we define as detection efficiency combined with the hit rate, the ALABAMA stated only the detection efficiency. We corrected this now in the revised version of the manuscript.

We appreciate the suggestion of adding a graph on the size-dependent detection efficiencies from the two single particle mass spectrometers and HINC, the latter with and without the PFPC. As our aim is to investigate which particle sizes are detected relatively, we normalized those measurements to the peak transmission efficiency from each instrument.

We updated the manuscript accordingly in lines 207 - 225, and include Fig. S1 in the appendix:

*The overall detection efficiency (combining the detection efficiency and the hit rate) of the LAAPTOF is between 0.01 (±0.01) % to 4.2 (±2.4) %, in the size range of 0.2 to 2 µm based on polystyrene latex particles (PSL). The highest overall detection efficiency is for 1µm and lowest for 2 µm (Shen et al., 2018). Note that such efficiency is also particle type dependent (Shen et al., 2018, 2019). More details on the LAAPTOF can be found in Gemayel et al. (2016), Reitz et al. (2016), and Shen et al. (2018, 2019). Details on the ALABAMA have been presented in Brands et al. (2011), Roth et al. (2016), Schmidt et al. (2017), and Clemen et al. (2020). The detection efficiency of the ALABAMA during this campaign was between 40% and 60% in the size range of 0.3 to 1.0 µm based on PSL particles. Up to a particle size of about 1.3 µm, the detection efficiency of the ALABAMA decreased to less than 30% and is estimated to be about 5 (±5) % for 2 µm. At the same time, the hit rate during those tests using PSL particles was lower, such that the overall detection efficiency for the ALABAMA was only between 1% and 16% in the size range from 0.3 to 1 µm. As those values are based on measurements using PSL particles, they can vary considerably during field applications; e.g., the ALABAMA hit rates were significantly higher than those of the PSL test measurements (which is attributed to particle charge effects during the nebulization of the PSL particles). In the light of our research objectives, focusing on the general trend of the aerosol particle composition, we therefore provide an overview of the size dependent overall detection efficiency from the LAAPTOF and the ALABAMA normalized to the maximum value measured, together with the normalized transmission efficiency from HINC (Fig. S1). From those normalized values it is visible that both SPMSs detect aerosol particles in the same size range, with a maximum between 0.5 and 1 µm, and therefore yield comparable information on the particle composition in this size range. HINC measures particles below 2 µm with a high efficiency which can have an impact on the comparison between the INP measurements from HINC and the results obtained from the ALABAMA and the LAAPTOF.*

[Figure]

*Figure S1: Normalized size-dependent detection efficiencies for aerosol particles in the ALABAMA and the LAAPTOF), and normalized transmission efficiency for HINC, and for HINC at the PFPC (based on calculations using the measured transmission efficiency with size-dependent enrichment from the PFPC (Gute et al., 2019)); the measurements were normalized to the maximum detection efficiencies of 4.2% (LAAPTOF), 16% (ALABAMA), 100% (HINC), and 1480% (HINC at PFPC).*

**It is really hard to determine what the Spearman's rho values actually mean. While it's a well-known equation, its main purpose is to detect if the relationship between two variables is monotonic. In this paper, it is used to define a correlation between INP concentrations and other aerosol measurements taken in parallel. It's a subtle distinction, and likely matters little if the rho values are very high (say > 0.8), but it becomes difficult to envision what a rho of 0.5 really means in this context. I suggest that the authors spend some time defining the equation and also plotting some rank correlations in the supplemental to help the reader envision how good / poor these rank correlations are.**

We agree with the reviewer and improve our description of the Spearman's rank correlation coefficient analysis. However, as the equation is widely known, we believe that it is dispensable to not present it in the manuscript but only to refer to the original publication by Spearman (1904).

At the end of section 2.2.1, we include now a description of the Spearman's rank correlation (lines 170 - 174)

*In order to investigate the relationship between $[INP]_{-31}$ and $n_s$ with meteorological and aerosol parameters, we use Spearman's rank correlation coefficient (Spearman, 1904). The Spearman's rank correlation determines to what extent two variables are monotonically related by applying a linear correlation analysis on the rank-ordered values of the parameters. As we would not necessarily await a linear relationship amongst parameters in atmospheric science, this test is well suited for our purposes. Examples for the correlation analysis are presented in Fig. S6.*

It is indeed difficult to define a „good" relationship between the investigated variables using correlation coefficients. For this we introduced the concept of „meaningful" relationships based on correlation coefficients larger than the mean + standard deviation. We agree with the reviewer that those values are difficult to envision, and therefore include now some examples for the aerosol particle chemistry parameters from the ALABAMA, presented in Fig. S6 in the revised supplement, panel b (previously Fig. S5).

[Figure]

*Figure S6: (Panel a) Spearman´s rank correlation coefficients for [INP]-31 (>LOD) with meteorological parameters, aerosol size distribution measurements, and aerosol particle chemistry determined with the WIBS (concentration all particles, fluorescent aerosol particles (FAP) and fluorescent biological aerosol particles (FBAP) within the size range of the instrument (> 0.5 μm)), the ALABAMA (mineral dust, sea spray, elemental carbon,* and sulfate *indicating* ions, according to the proxies defined in Figs. 6 and 7, *see section 3.3, and with the aethalometer*

*(elemental black carbon, eBC); (panel b) examples for the visualization of Spearman's rank correlation for [INP]$_{-31}$ with the ambient temperature, particle concentration > 0.5 μm, and aerosol particle chemistry parameters determined with the ALABAMA (values in brackets reflect the value of the correlation coefficient) .*

**Likewise, I'm not sure that taking the mean ± std. dev. of the rank correlations really defines what m/z have "significant" correlations. It makes the assumption that, in general, peaks in single-particle mass spec. are not correlated with INP concentrations. This assumption was not rigorously tested, at least in this paper, and I see no good reason why it would be true. For example, Mineral Dust spectra contain many relatively unique peaks; ostensibly, these peaks will correlate with INP concentrations.**

We would like to point out that we did not make any assumptions before correlating the presence of ions with the INP concentration. On the contrary, we used the approach of correlating the m/z signals to perform an analysis that was initially independent of particle types, giving us an overview of the correlations of individual ions with INP concentration. Based on this analysis, the meaningful correlators were then assigned to potential particle types. It is concluded that meaningful correlators are ions that are mainly found in the mass spectra of INPs active at -31 °C or that have a comparable temporal occurrence to INPs active at -31 °C. Mineral dust spectra will correlate with INP if the mineral dust particles indeed acted as INP. However, coatings may inhibit the INP properties of mineral dust (see e.g. Sullivan et al., 2010). Our approach was to test whether we find correlations at all and whether we derive INP properties from these correlations.

We agree that the reviewer is right in that we use the term „significant" incorrectly here. By only investigating ranked correlation coefficients larger than mean + standard deviation we are aiming at identifying relationships between the variables which are above a „background noise". We change our wording accordingly in the manuscript in lines 276 - 277:

*Only ions whose $r^2$ values were greater than 1σ above the mean value were selected as meaningful correlators (see Supplement Fig. S3) to [INP]$_{-31}$ or n$_s$.*

And in lines 463 - 464:

*Although $r^2$ was chosen to determine which ions are meaningfully correlated with [INP]$_{-31}$ and n$_s$, we consider r to also identify negative correlations.*

**One of the most troubling aspects of this paper is that some of its results contradict themselves. For example, Figure 5 indicates that sea-spray aerosol (SSA) are correlated with INP; however, Figure 7 indicates that the opposite is true. One reason for this is that many peaks are not unique to a single particle type. Thus, trying to attribute a peak to single particle type as done in Figure 5 may give erroneous results. For example, m/z 60 is defined as mineral dust (SiO+2 for LAAPTOF, but elemental carbon (C+5) for ALABAMA. I suspect that it shows up in each particle type for both single-particle mass specs.**

We want to point out that for the LAAPTOF, we refer to the negative ion at m/z 60 (SiO$_2^-$) as a marker for mineral dust, whereas for the ALABAMA, the positive ion at m/z 60 is often observed

in connection with $C_5^+$ and therefore a marker for elemental carbon, especially when all other $C_n^+$ ions are also found in a particle spectrum. The list of marker peaks in Fig. 5 originates from measurements of reference or test materials (Schmidt et al., 2017; Shen et al., 2018), with both mass spectrometers and represents the most likely identifications of ions. In order to improve our description of the result interpretation, we add the following text to the manuscript (lines 412 - 417):

*The interpretation of particle components and particles types was achieved by comparison with existing reference mass spectra from both mass spectrometers (see Figs. S8 and S9, panel a), so we consider the ions listed below and their assignment to the particle types as quite likely, even considering that potentially multiple ions can be assigned to a single m/z ratio. Furthermore, it should be noted that the ions presented in this chapter could also be assigned to multiple particle types, with an ion being a meaningful correlator only if the major representatives of the associated particle types also have approximately meaningful correlations with $[INP]_{-31}$ and $n_s$.*

We do not agree that Figs. 5 and Fig 7 contradict each other. The "sea spray proxy" shown in Fig. 7, constructed from the marker ions listed in Fig 5, has a correlation coefficient of 0.62 (as shown now in the updated Fig. S6) which we consider to depict a meaningful relationship between the sea spray particle proxy and the $[INP]_{-31}$. Furthermore, we want to point out that Fig. 7 contains more data than used for Fig.5. For the correlation coefficients between the ions and INP concentrations shown in Fig. 5, we had to restrict the analysis to time periods where the ALABAMA, the LAAPTOF and HINC measured simultaneously. In Fig. 7, we used the whole time series available from the ALABAMA and the LAAPTOF to inspect the connection between the mass spectrometer time series and the FLEXPART results. Still, the simulated FLEXPART time series of marine surfaces does not show good temporal agreement with those measured values. This discrepancy can be attributed in part to the fact that all marine surfaces were considered in the simulations, and that they were not restricted to shorelines with increased wave breaking activity, which is considered a main aerosol emission source. Furthermore, the possibility that the sea spray particle type is not only constrained to marine regions but may also be attributed to dry saline lake beds in the deserts is already discussed in Section 3.3.7.

Moreover, typically, not only one but several m/z values, indicating of a certain aerosol type, is observed to be amongst the best correlators; e.g., different m/z values indicating dust particles ($SiO^+$, $SiO_3^-$, $HSiO_3^-$, $CaO^+$) appear as top correlators in Fig. 5.

Minor Comments

**Some Figures are extremely hard to read in print format. I cannot read the axes in Figure 4, some text and the "markers" in part C of Figure 5, and the legends in Figures 9 and 11.**

We agree and improve the readability of Figs. 4, 5, 9 and 11.

**Line 155: A detailed description of how the concentration factor needs to be added here. To say that you increase aerosol >100 by a factor of 20 is greatly simplifying what is happening.**

We extend the description of the INP measurements at the PFPC now (lines 151 - 158):

*To achieve better measurement statistics due to the naturally low [INP] in the free troposphere, an aerosol particle concentrator (the Portable Fine Particle Concentrator (PFPC; Gute et al., 2019) was deployed upstream of HINC during the field campaign, allowing an enrichment in aerosol particles > 0.1 µm. The enrichment is thereby size-dependent due to the working principle of the PFPC, with an enrichment factor of ~ 10 at particle sizes of 0.3 µm, a maximum enrichment of ~20 for particles > 0.75 µm (Gute et al., 2019). The INP enrichment factors were determined by consecutive measurements on and off the concentrator, and showed a large variability between values of 1 and 23, reflecting the variability in the size of the present INP population (see Lacher et al., 2018a, for a more detailed description of this setup).*

**Line 160: The caveats of converting ambient [INP] to $n_s$ should be addressed here. By definition, $n_s$ assumes that ice nucleation is deterministic, but it has been shown that this is not necessarily true for aerosol populations with a wide variety of ice-active site densities.**

We agree with the reviewer's statement and include now the following discussion in the manuscript (lines 166 – 169) :

*The concept of $n_s$ is based on the assumption of a uniform composition of the investigated aerosol sample, and assumes that the temperature dependence of ice nucleation is more important than the time dependence, which is therefore neglected (e.g., Welti et al., 2012).*

**Line 245: You are biased by the mass spec. detection efficiencies. Was there any attempt at normalization to the optical particle counters?**

We did not normalize the mass spec results to optical counter measurements, because firstly, we are looking at correlations of individual ions and not at absolute particle number concentrations, and secondly, the transmission and detection efficiency curves of both mass spectrometers (shown above) are similar, so we can safely assume that we analyzed similar particle populations with both mass spectrometers. In order to use single particle mass spectrometry results in a quantitative manner, you need to have very precise size distribution measurements from the particles detected with the single particle mass spectrometers (e.g., Froyd et al., 2019), which was not intended during the campaign.

**Line 469: This statement is incorrect for two reasons. One -a large signal in one particle could greatly skew the average because the signal spans several orders of magnitude and a linear average was applied. That is one INP disproportionately affecting the signal average. Two-a single peak in a mass spectrum may qualitatively scale with abundance**

**of that element or fragment, but it likely does not scale with the abundance of a "substance" or ice-active site.**

We agree with the reviewer. This issue is handled by using many particles and ions for the analysis, i.e. we reduce the effect of individual outliers by statistics. The abundance of the fragment is generally related with the abundance of the parent substance(s), since this is their origin. A better correlation between the normalized intensity of a certain peak and the INP concentration could suggest that the abundance of the corresponding substance(s) play a role, in spite of a potentially non-linear relationship. This is one of the reasons why we used Spearman rank correlation. In addition, we are focusing on the pattern of ion signatures (several characteristic peaks) rather than a single peak. This strengthens the conclusion we have made.

**Line 537: I believe that dust particles fluoresce slightly in the WIBS. This is one reason why the FBAP thresholds are so strict. This should be mentioned here-or it should be mentioned why dust may not be suspected.**

We already mention this in section 2.4, lines 298 - 299:

*It should be pointed out that fluorescence in any of the 3 channels can be impacted by non-biological particles such as dust;…*

and in section 3.4, lines 581 - 582:

*As depicted in Fig. S10, panel b, mineral dust particles mainly occur in the size ranges > 0.5 µm, and they can also show fluorescence.*

**Line 626: I'm not quite sure how you arrive at the conclusion that 70% of EC particles also contain mineral dust. From general aerosol knowledge, this seems like a vast overestimation. Thus, more details are need to support this statement.**

This was a first-order estimate based on measurements of single particle mass spectra. However, the uncertainties of this statement are too great, as it is difficult to precisely define EC containing particles. Therefore, based on the reviewer comment, we decided to delete this statement.

Technical Comments

**Line 42: Are metallic particles and biological particles each 10% or are they 10% together?**

Each are 10%, corrected in line 40.

*…and also biological and metallic particles are found to a smaller extent (~10% each).*

**Line 60: I would not say that INP concentrations are entirely "unconstrained." Plenty of papers show a reasonable range of ambient INP concentrations depending on the air mass.**

Agreed and changed (lines 55 - 56):

*Despite their importance, the knowledge about the abundance and nature of INPs in the atmosphere still needs to be improved,…*

**Line 116: Is there an estimate of the transport time from the MBL to JFJ?**

This is indeed an interesting question. We analysed the FLEXPART simulations in more depth and tracked when, where and how intense the contact to the marine boundary layer occurred for air masses sampled at Jungfraujoch station. We considered the residence time of model particles below 100 m above sea level and recorded their travel time after leaving the marine boundary layer. In general, we find that travel times from those marine boundary layer areas to Jungfraujoch can vary strongly between approximately 24 hours to over 96 hours. Interestingly, at times the single particle mass spectrometers recorded a stronger signal of sea spray particles (February 2 - 5) the travel times were rather short (12 to 24 hours) and most of the contact to the marine boundary layer occurred over the Western Mediterranean. The short travel time is in line with a reduced impact of sedimentation losses, wet removal, and mixing/dilution with other air masses of different origins.

We include now the following statement to that effect in the revised version of the manuscipt in section 2.6 (lines 343 - 346)

*Moreover, to improve the understanding of travel times from the marine boundary layer to JFJ, the FLEXPART simulations are tracked for the location, time, and intensity of the marine boundary contact for air masses sampled at JFJ. The residence time of model particles below 100 m above sea level is considered and their travel time after leaving the marine boundary layer is recorded.*

And updated the statement in section 3.3.7 (lines 542 - 560)

*However, the time series of the sea spray proxies from the mass spectra do not match the FLEXPART emission footprint sensitivities for open ocean and sea surfaces (Fig. 7, panel a). Also, the [INP]-31 time series does not match the open water source regions from FLEXPART. Interestingly, the calculated air mass travel times from FLEXPART between the marine boundary layer and the measurement site reveal similarities with the sea spray proxies. For example, the period when the single-particle mass spectrometers recorded a stronger signal from sea spray particles (2 to 5 February) coincides with comparatively short travel times of the air masses to the JFJ (12 to 24 hours), mainly originating from the marine boundary layer over the western Mediterranean Sea (Fig. S11). The short travel time is in line with a reduced impact of sedimentation losses, wet removal, and mixing/dilution with other air masses of different origins. The FLEXPART analysis shows that travel times from the marine boundary layer to JFJ, can vary between approximately 24 hours and more than 96 hours. However, we should point out that the FLEXPART model is not able to separate coastal regions from open*

*ocean for this analysis, such that the open water classification used may not be the best way to represent sea spray source regions which tend to occur along shorelines during wave breaking and as function of windspeed (e.g. Vergara-Temprado et al., 2017).*

*Apart from this, both the sea spray proxie and the [INP]-31 time series show a similar temporal evolution as the dust source region emission sensitivities (Fig. 6). A possible explanation for this might be found in the dry saline lake beds in the deserts have been identified as source regions for salt particles (Prospero et al., 2002). For example, Formenti et al. (2003), observed that airborne particles originating from the Sahara that were sampled off the North African coast were a mixture of aluminium-silicate based minerals and NaCl bearing salts. This may well explain our observed correlation between dust particles, NaCl containing particles and INP.*

and add the figure below to the supplementary information as Fig. S11

[Figure]

*Figure S11: Residence time of FLEXPART-simulated particles in the marine boundary layer as function of arrival time at Jungfraujoch (x-axis) and travel time to the site (y-axis). Units reflect default output of FLEXPART for backward simulations, which is residence time divided by air density.*

**Line 141: I think it should be "[INP] are naturally low ..."**

It should be „… the INP concentration is naturally low, ...", corrected in line 138:

*… the [INP] is naturally low, …*

**Line 141: This sentence could be split into two sentences. The phrase "but is generally higher ..." is difficult to follow because there are many potential subjects -e.g., nucleation temperature, detection limit, [INP], etc.**

Agreed and changed in lines 137 - 139:

*"A nucleation temperature of -31 °C was chosen in order to avoid measurements below the detection limit of the instrument. This is crucial at a remote location as JFJ where the [INP] is naturally low, but is generally higher at colder nucleation temperatures."*

**Line 190: Should it be "into" instead of "onto?"**

We rephrased the sentence to (lines 188 - 189):

*... the resulting suspension is then pipetted into 80 wells of PCR plates, with each well having a volume of 50 μL,..*

**Line 200: Should it be "decreases" instead of "decreased?"**

We decided to use the past tense here, as those settings were valid for this specific field campaign. The ALABAMA was recently subject to improvement as presented in Clemen et al. (2020), increasing the detection efficiency.

**Line 206: Should it be "mJ / pulse?"**

Correct, we changed the sentence to (lines 229 - 230):

*The laser energy per pulse used during this campaign ranged between 3 mJ and 4 mJ for the LAAPTOF and between 7.2 mJ and 9 mJ for the ALABAMA.*

**Line 289: What is the typical size range of ice crystals at JFJ?**

Ice crystals can span a wide range of sizes from below to a few micrometers (freshly nucleated ice crystals) to several tens and hundreds of micrometers (e.g., Korolev et al., 2017). This depends on the temperature and supersaturation conditions in the clouds and the age of the ice crystals. This is also the size range of ice crystals measured during field measurements at the Jungfraujoch station (Henneberger et al., 2013).

**Line 343: There is a lot of information in Figure 2, so it would be instructive to highlight these periods in Figure 2.**

Agreed and changed.

**Line 353: This reference to the Kammermann paper is not really supported by Figure 9.**

We agree with the reviewer that most of the mineral dust IPRs depicted in Fig. 9 are submicron in size. However, this is not in contradiction to the statement that supermicron particles are indicative for dust particles; the Ice-CVI only samples the subset of particles being ice-active during in-cloud conditions, and thus is not representative of the aerosol particle size distribution in ambient air.

**Line 483: Please move the legends in Figures 6 and 7, they are obstructing one of the high INP periods.**

Agreed and changed.

**Line 563: More secondary ice from your calculations, right? The way this sentence reads, it sounds like it's not an inferred measurement.**

Yes, that is correct. We changed the sentence into (lines 603 - 605):

*This assumption is further supported by the fact that on average the cloud periods interpreted as being stronger influenced by secondary ice contribution have significantly higher IPR number concentrations.*

References

Clemen, H.-C., Schneider, J., Klimach, T., Helleis, F., Köllner, F., Hünig, A., Rubach, F., Mertes, S., Wex, H., Stratmann, F., Welti, A., Kohl, R., Frank, F., and Borrmann, S.: Optimizing the detection, ablation, and ion extraction efficiency of a single-particle laser ablation mass spectrometer for application in environments with low aerosol particle concentrations, Atmos. Meas. Tech., 13, 5923-5953, 10.5194/amt-13-5923-2020, 2020.

Froyd, K. D., Murphy, D. M., Brock, C. A., Campuzano-Jost, P., Dibb, J. E., Jimenez, J. L., Kupc, A., Middlebrook, A. M., Schill, G. P., Thornhill, K. L., Williamson, C. J., Wilson, J. C., and Ziemba, L. D.: A new method to quantify mineral dust and other aerosol species from aircraft platforms using single-particle mass spectrometry, Atmos. Meas. Tech., 12, 6209-6239, 10.5194/amt-12-6209-2019, 2019.

Henneberger, J., Fugal, J. P., Stetzer, O., and Lohmann, U.: HOLIMO II: a digital holographic instrument for ground-based in situ observations of microphysical properties of mixed-phase clouds, Atmos. Meas. Tech., 6, 2975-2987, 10.5194/amt-6-2975-2013, 2013.

Korolev, A., McFarquhar, G., Field, P. R., Franklin, C., Lawson, P., Wang, Z., Williams, E., Abel, S. J., Axisa, D., Borrmann, S., Crosier, J., Fugal, J., Krämer, M., Lohmann, U., Schlenczek, O., Schnaiter, M., and Wendisch, M.: Mixed-Phase Clouds: Progress and Challenges, Meteorol. Monogr., 58, 5.1-5.50, 10.1175/amsmonographs-d-17-0001.1, 2017.

Rolph, G., Stein, A., and Stunder, B.: Real-time Environmental Applications and Display sYstem: READY, Environ. Modell. Software, 95, 210-228, https://doi.org/10.1016/j.envsoft.2017.06.025, 2017.

Shen, X., Ramisetty, R., Mohr, C., Huang, W., Leisner, T., and Saathoff, H.: Laser ablation aerosol particle time-of-flight mass spectrometer (LAAPTOF): performance, reference spectra and classification of atmospheric samples, Atmos. Meas. Tech., 11, 2325-2343, 10.5194/amt-11-2325-2018, 2018.

Schmidt, S., Schneider, J., Klimach, T., Mertes, S., Schenk, L. P., Kupiszewski, P., Curtius, J., and Borrmann, S.: Online single particle analysis of ice particle residuals from mountain-top mixed-phase clouds using laboratory 1145 derived particle type assignment, Atmos. Chem. Phys., 17, 575-594, 10.5194/acp-17-575-2017, 2017.

Schwikowski, M., Seibert, P., Baltensperger, U., and Gaggeler, H. W.: A study of an outstanding Saharan dust event at the high-alpine site Jungfraujoch, Switzerland, Atmospheric Environment, 29, 1829-1842, https://doi.org/10.1016/1352-2310(95)00060-C, 1995.

Spearman, C.: The Proof and Measurement of Association between Two Things, The American Journal of Psychology, 15, 72-101, 10.2307/1412159, 1904.

Stein, A. F., Draxler, R. R., Rolph, G. D., Stunder, B. J. B., Cohen, M. D., and Ngan, F.: NOAA's HYSPLIT Atmospheric Transport and Dispersion Modeling System, Bull. Am. Meteorol. Soc. , 96, 2059-2077, 10.1175/bams-d-14-00110.1, 2015.

Sullivan, R. C., Petters, M. D., DeMott, P. J., Kreidenweis, S. M., Wex, H., Niedermeier, D., Hartmann, S., Clauss, T., Stratmann, F., Reitz, P., Schneider, J., and Sierau, B.: Irreversible loss of ice nucleation active sites in mineral dust particles caused by sulphuric acid condensation, Atmos. Chem. Phys., 10, 11471–11487, https://doi.org/10.5194/acp-10-11471-2010, 2010.

---

## Author Response (AR2)

**Author comment in response to the comments provided by the Editor and Referee #2, second round of revision**

We thank the Editor and Referee #2 for their effort in reading and commenting our manuscript carefully again and giving helpful feedback. In the following, we repeat the **referee's comments in bold typeface**, and give point-by-point answers in normal typeface; extracts from the *original manuscript are presented in red italic*, and from the *revised manuscript in blue italic*. Line numbers are referring to the updated manuscript version.

General Comments:

**Dear Authors,**

**I received the reviews for your revised manuscript. Reviewer #2 still has some criticism how single mass spectrometer peaks are related to particle types (given below). As it is presented in the manuscript, I agree that this can be confusing and ambiguous.**

**Clearly, a specific particle type cannot be identified by a single mass peak. Looking at Schmidt et al. (2017) and Shen et al. (2018), several positive and negative peaks are applied to identify the particle type.**

**However, the text on page 7 (lines 258-272) states in several places that the focus is on single ions and either the fraction or relative intensity of a single ion is used for particle identification. This is somehow confusing.**

**In Fig. 5, bottom (c), several mass peaks are indicated for identification of the particle type. However, the correlation analysis shown in (a) and (b) focused on single ions (at least it appears like this)? As pointed out by the reviewer, a single mass peak is not sufficient, sometimes present in several different particle types, and thus renders analysis ambiguous.**

**Could you please elaborate on this point?**

**With kindest regards,**

**Daniel Knopf**

**Reviewer #2:**

**Overall, the authors have addressed my comments; however, I still disagree with their statements regarding Figure 5. Specifically, the authors have not addressed that single peaks often do not correlate to specific particle types. For example, in ALABAMA negative spectra, m/z 12 can be found in almost all of the particle types in Figure S9. The authors, however, interpret an ALABAMA anion with a high m/z 12 r_pos as "carbon," which the authors sometimes interpret as elemental carbon. Similarly, in the LAAPTOF positive spectra in Figure S10, many particle types contain large m/z 44 and 46 peaks, but Figure 5 suggests that these are mineral dust and sea spray, respectively. This reviewer suggests that this ambiguity makes this analysis much less useful that the authors suggest.**

**Perhaps more useful to this reviewer are the plots with black and pink markers in Figure S9. Here, the authors can correlate known particle types with INP, not just single peaks. This reviewer strongly suggests that the analysis in Figure 5 be revisited, and, if it remains in the paper, that the authors build a stronger case for assigning particles types to single m/z values (i.e., assigning colors to each m/z), or remove that analysis altogether.**

We agree with the editor and the reviewer that single ion peaks cannot be used to identify specific particle types. We would like to point out that this is not the way we analyzed our data and realize that our explanation was not clear enough. We therefore changed the way our analysis is presented in the manuscript. Now, we first present the m/z correlation analysis (from previous Fig. 5, panel a and b) and turn this into Tab. 1. The assignment of ions to the m/z values and the use of selected ions as markers for particle types (formerly Fig. 5, panel c) is now presented separately in a new Tab. S1. As such we separate the discussion about the single ion correlation analysis and the ion assignment to m/z values and to particle types. From the new Tab. S1 it becomes clear that we are aware of the different possible ions contributing to one m/z and the occurrence of certain ions in several particle types.

*Table1: Ranked correlation coefficients for [INP]$_{-31}$ and respective $n_s$ (both determined > LOD) with m/z values for LAAPTOF (panel a) and ALABAMA (panel b), for method 1 and 2, respectively. The assignment of potential ions to m/z values can be found in the supporting information (Table S1).*

(a) LAAPTOF

**Method 1 cations / Method 1 anions**

| Rank | m/z | r([INP]$_{-31}$) | m/z | r($n_s$) | m/z | r([INP]$_{-31}$) | m/z | r($n_s$) | m/z | r([INP]$_{-31}$) | m/z | r($n_s$) | m/z | r([INP]$_{-31}$) | m/z | r($n_s$) |
|---|---|---|---|---|---|---|---|---|---|---|---|---|---|---|---|---|
| | | r_pos | | | | r_neg | | | | r_pos | | | | r_neg | | |
| 1 | 46 | 0.51 | 56 | 0.36 | 29 | -0.42 | 13 | -0.30 | 60 | 0.59 | 60 | 0.39 | | | | |
| 2 | 45 | 0.50 | 46 | 0.34 | | | | | 76 | 0.55 | 76 | 0.36 | | | | |
| 3 | 44 | 0.47 | 96 | 0.34 | | | | | 77 | 0.53 | 35 | 0.35 | | | | |
| 4 | 57 | 0.47 | 74 | 0.32 | | | | | 37 | 0.51 | 77 | 0.35 | | | | |
| 5 | 56 | 0.46 | 57 | 0.31 | | | | | 35 | 0.49 | 72 | 0.32 | | | | |
| 6 | 74 | 0.44 | 45 | 0.30 | | | | | 59 | 0.47 | 37 | 0.32 | | | | |
| 7 | 96 | 0.43 | 44 | 0.30 | | | | | 120 | 0.46 | 120 | 0.32 | | | | |
| 8 | 58 | 0.42 | 41 | 0.29 | | | | | 50 | 0.45 | 50 | 0.32 | | | | |
| 9 | 75 | 0.42 | | | | | | | 43 | 0.44 | 16 | 0.31 | | | | |
| 10 | 76 | 0.41 | | | | | | | 28 | 0.44 | | | | | | |
| 11 | 81 | 0.37 | | | | | | | 16 | 0.44 | | | | | | |
| 12 | 82 | 0.36 | | | | | | | 17 | 0.43 | | | | | | |
| 13 | 83 | 0.35 | | | | | | | 72 | 0.43 | | | | | | |
| 14 | | | | | | | | | 42 | 0.43 | | | | | | |
| 15 | | | | | | | | | 36 | 0.41 | | | | | | |

**Method 2 cations / Method 2 anions**

| Rank | m/z | r([INP]$_{-31}$) | m/z | r($n_s$) | m/z | r([INP]$_{-31}$) | m/z | r($n_s$) | m/z | r([INP]$_{-31}$) | m/z | r($n_s$) | m/z | r([INP]$_{-31}$) | m/z | r($n_s$) |
|---|---|---|---|---|---|---|---|---|---|---|---|---|---|---|---|---|
| | | r_pos | | | | r_neg | | | | r_pos | | | | r_neg | | |
| 1 | 44 | 0.57 | 44 | 0.37 | 13 | -0.37 | 30 | -0.33 | 60 | 0.60 | 60 | 0.38 | | | | |
| 2 | 46 | 0.50 | 56 | 0.33 | | | | | 76 | 0.55 | 35 | 0.36 | | | | |
| 3 | 45 | 0.47 | 46 | 0.33 | | | | | 77 | 0.54 | 76 | 0.35 | | | | |
| 4 | 56 | 0.46 | 96 | 0.32 | | | | | 37 | 0.52 | 77 | 0.34 | | | | |
| 5 | 96 | 0.44 | 41 | 0.32 | | | | | 35 | 0.52 | 72 | 0.32 | | | | |
| 6 | 75 | 0.43 | 74 | 0.30 | | | | | 36 | 0.47 | 120 | 0.31 | | | | |
| 7 | 76 | 0.43 | | | | | | | 120 | 0.47 | 73 | 0.31 | | | | |
| 8 | 74 | 0.43 | | | | | | | 59 | 0.46 | 19 | 0.30 | | | | |
| 9 | 57 | 0.41 | | | | | | | 50 | 0.44 | | | | | | |
| 10 | 81 | 0.41 | | | | | | | 72 | 0.44 | | | | | | |
| 11 | 58 | 0.40 | | | | | | | 28 | 0.42 | | | | | | |
| 12 | 82 | 0.38 | | | | | | | 43 | 0.42 | | | | | | |
| 13 | 40 | 0.37 | | | | | | | 29 | 0.42 | | | | | | |
| 14 | 83 | 0.37 | | | | | | | 73 | 0.409 | | | | | | |

(b) ALABAMA

**Method 1 cations**

| | r_pos | | | | r_neg | | | |
|---|---|---|---|---|---|---|---|---|
| Rank | m/z | r([INP]-31) | m/z | r(ns) | m/z | r([INP]-31) | m/z | r(ns) |
| 1 | 165 | 0.47 | 78 | 0.44 | | | | |
| 2 | 78 | 0.46 | 144 | 0.42 | | | | |
| 3 | 7 | 0.44 | 165 | 0.41 | | | | |
| 4 | 35 | 0.43 | 62 | 0.39 | | | | |
| 5 | 62 | 0.41 | 63 | 0.37 | | | | |
| 6 | 22 | 0.40 | 222 | 0.37 | | | | |
| 7 | 23 | 0.39 | 224 | 0.37 | | | | |
| 8 | 133 | 0.38 | 132 | 0.36 | | | | |
| 9 | 164 | 0.38 | 23 | 0.36 | | | | |
| 10 | 144 | 0.37 | 151 | 0.36 | | | | |
| 11 | 224 | 0.35 | 108 | 0.35 | | | | |
| 12 | 132 | 0.35 | 202 | 0.35 | | | | |
| 13 | 222 | 0.34 | 217 | 0.35 | | | | |
| 14 | 118 | 0.34 | 240 | 0.34 | | | | |
| 15 | 20 | 0.34 | 181 | 0.34 | | | | |
| 16 | 151 | 0.34 | 118 | 0.34 | | | | |
| 17 | | | 180 | 0.34 | | | | |
| 18 | | | 223 | 0.33 | | | | |
| 19 | | | 231 | 0.33 | | | | |
| 20 | | | 213 | 0.33 | | | | |
| 21 | | | 133 | 0.32 | | | | |

**Method 1 anions**

| | r_pos | | | | r_neg | | | |
|---|---|---|---|---|---|---|---|---|
| Rank | m/z | r([INP]-31) | m/z | r(ns) | m/z | r([INP]-31) | m/z | r(ns) |
| 1 | 12 | 0.48 | 42 | 0.47 | 141 | -0.53 | 97 | -0.39 |
| 2 | 18 | 0.45 | 43 | 0.43 | 99 | -0.51 | 195 | -0.37 |
| 3 | 37 | 0.44 | 12 | 0.42 | 97 | -0.49 | | |
| 4 | 35 | 0.44 | 27 | 0.40 | 155 | -0.46 | | |
| 5 | 4 | 0.44 | 37 | 0.39 | 96 | -0.45 | | |
| 6 | 9 | 0.43 | 35 | 0.37 | 217 | -0.42 | | |
| 7 | 23 | 0.43 | 9 | 0.37 | 115 | -0.40 | | |
| 8 | 34 | 0.43 | 40 | 0.34 | 101 | -0.40 | | |
| 9 | 50 | 0.39 | | | | | | |
| 10 | 3 | 0.39 | | | | | | |

**Method 2 cations**

| | r_pos | | | | r_neg | | | |
|---|---|---|---|---|---|---|---|---|
| Rank | m/z | r(INP) | m/z | r(ns) | m/z | r(INP) | m/z | r(ns) |
| 1 | 60 | 0.61 | 60 | 0.55 | | | | |
| 2 | 51 | 0.61 | 132 | 0.52 | | | | |
| 3 | 24 | 0.61 | 39 | 0.47 | | | | |
| 4 | 12 | 0.57 | 78 | 0.44 | | | | |
| 5 | 62 | 0.57 | 63 | 0.44 | | | | |
| 6 | 36 | 0.57 | 84 | 0.44 | | | | |
| 7 | 23 | 0.55 | 108 | 0.43 | | | | |
| 8 | 40 | 0.55 | 62 | 0.43 | | | | |
| 9 | 78 | 0.54 | 36 | 0.42 | | | | |
| 10 | 165 | 0.54 | 120 | 0.41 | | | | |
| 11 | 84 | 0.53 | 240 | 0.39 | | | | |
| 12 | 63 | 0.52 | 23 | 0.38 | | | | |
| 13 | 132 | 0.52 | 91 | 0.38 | | | | |
| 14 | 35 | 0.50 | 165 | 0.38 | | | | |
| 15 | 67 | 0.49 | 24 | 0.38 | | | | |
| 16 | 22 | 0.46 | 76 | 0.38 | | | | |
| 17 | 133 | 0.45 | 41 | 0.37 | | | | |
| 18 | 17 | 0.43 | 138 | 0.37 | | | | |
| 19 | 120 | 0.43 | | | | | | |

**Method 2 anions**

| | r_pos | | | | r_neg | | | |
|---|---|---|---|---|---|---|---|---|
| Rank | m/z | r(INP) | m/z | r(ns) | m/z | r(INP) | m/z | r(ns) |
| 1 | 76 | 0.54 | 42 | 0.40 | 155 | -0.51 | 97 | -0.36 |
| 2 | 12 | 0.50 | 76 | 0.39 | 97 | -0.51 | 177 | -0.35 |
| 3 | 35 | 0.49 | 12 | 0.38 | 141 | -0.50 | 195 | -0.34 |
| 4 | 37 | 0.46 | 9 | 0.37 | 106 | -0.44 | | |
| 5 | 19 | 0.45 | 43 | 0.37 | 99 | -0.44 | | |
| 6 | 32 | 0.44 | 35 | 0.34 | 233 | -0.41 | | |
| 7 | 18 | 0.44 | 78 | 0.34 | | | | |
| 8 | 4 | 0.44 | 37 | 0.33 | | | | |
| 9 | 77 | 0.44 | 19 | 0.33 | | | | |
| 10 | 9 | 0.43 | | | | | | |

Table S1: Possible ions contributing to the best correlating m/z values from Tab. 1, along with their occurrence in different particle types and the use of selected ions as markers for certain particle types for ALABAMA and LAAPTOF, for cations (panel a) and anions (panel b).

| (a) cations m/z | potential ions | mineral dust | sea spray | EC | sulfate containing | ALA | LAAP |
|---|---|---|---|---|---|---|---|
| 7 | $Li^+$ | $Li^+$ | - | - | - | | |
| 12 | $C_1^+$, $Mg^{2+}$ | $Mg^{2+}$ | $Mg^{2+}$ | $C_1^+$ | $C_1^+$ | gray | |
| 13 | $C_1H^+$ | - | - | - | - | | |
| 17 | $NH_3^+$ | - | - | - | - | | |
| 20 | $^{40}Ca^{2+}$ | $^{40}Ca^{2+}$ | $^{40}Ca^{2+}$ | - | - | | |
| 22 | $^{44}Ca^{2+}$ | $^{44}Ca^{2+}$ | $^{44}Ca^{2+}$ | - | - | | |
| 23 | $Na^+$ | $Na^+$ | $Na^+$ | $Na^+$ | $Na^+$ | blue | |
| 24 | $C_2^+$, $Mg^+$ | $Mg^+$ | $Mg^+$ | $C_2^+$ | $C_2^+$ | | |
| 29 | $C_2H_5^+$, $CHO^+$, $CH_2NH^+$ | - | - | - | - | | |
| 35 | $NH_4NH_3^+$ | - | - | - | - | | |
| 36 | $C_3^+$ | - | - | $C_3^+$ | - | gray | |
| 39 | $^{39}K^+$, $C_3H_3^+$ | $^{39}K^+$ | $^{39}K^+$ | $^{39}K^+$ | $^{39}K^+$ | | |
| 40 | $^{40}Ca^+$, $MgO^+$ | $^{40}Ca^+$ | $^{40}Ca^+$, $MgO^+$ | - | - | orange | orange |
| 41 | $^{41}K^+$, $MgOH^+$, $Na(H_2O)^+$ | $^{41}K^+$ | $^{41}K^+$ | $^{41}K^+$ | $^{41}K^+$ | | |
| 44 | $^{44}Ca^+$, $SiO^+$ | $^{44}Ca^+$, $SiO^+$ | $^{44}Ca^+$ | - | - | | orange |
| 45 | $COOH^+$, $CHS^+$, $CH_3CHOH^+$, $CH_3OCH_2^+$ | - | - | - | - | | |
| 46 | $Na_2^+$, $(CH_3)_2NH_2^+$, $CH_2S^+$ | - | $Na_2^+$ | - | - | | blue |
| 51 | $V^+$, $C_4H_3^+$ | - | - | - | - | | |
| 56 | $CaO^+$, $Fe^+$, $Si_2^+$, $KOH^+$ | $CaO^+$, $Fe^+$, $Si_2^+$ | $CaO^+$ | - | $KOH^+$ | orange | |
| 57 | $CaOH^+$, $C_4H_9^+$, $C_2O_2H^+$, $C_3H_5O^+$ | $CaOH^+$ | $CaOH^+$ | - | - | orange | orange |
| 58 | $(CH_2)C_2H_5NH^+$ | - | - | - | - | | |
| 60 | $C_5^+$, $AlO_2H^+$ | - | - | $C_5^+$ | - | gray | |
| 62 | $Na_2O^+$, $(CH_3)NHOH$ | - | $Na_2O^+$ | - | - | blue | |
| 63 | $Na_2OH^+$, $C_5H_3^+$, $Cu^+$ | - | $Na_2OH^+$ | - | - | blue | |
| 67 | $VO^+$, $CaAl^+$, $C_3H_3N_2^+$, $NaSiO^+$ | - | - | - | - | | |
| 74 | $C_3H_6O_2^+$, $(C_2H_5)_2NH_2^+$, $N(CH_3)_4^+$ | - | - | - | - | | |
| 75 | $CaCl^+$ | $CaCl^+$ | - | - | - | | orange |
| 76 | $(CH_3)_3NOH^+$ | - | - | - | - | | |
| 78 | $K_2^+$ | - | $K_2^+$ | - | - | blue | |
| 81 | $Na_2{}^{35}Cl^+$ | - | $Na_2{}^{35}Cl^+$ | - | - | | blue |
| 82 | $CaCNO^+$, $HBr^+$ | - | - | - | - | | |
| 83 | $Na_2{}^{37}Cl^+$, $VO_2^+$, $CaAlO^+$ | $CaAlO^+$ | $Na_2{}^{37}Cl^+$ | - | - | | blue |
| 84 | $C_7^+$, $VHO_2^+$, $Si_3^+$, $C_5NH_{10}^+$ | - | - | $C_7^+$ | - | gray | |
| 91 | $C_7H_7^+$, $C_6H_3O^+$, $C_4H_8Cl^+$, $(CH_3)_2NHNO_2^+$ | - | - | - | - | | |
| 96 | $Ca_2O^+$, $C_8^+$ | $Ca_2O^+$ | - | $C_8^+$ | - | gray | orange |
| 108 | $C_9^+$, $Na_2NO_3^+$ | - | $Na_2NO_3^+$ | $C_9^+$ | - | gray | |
| 118 | $(C_2H_5)_3NOH^+$ | - | - | - | - | | |
| 120 | $C_{10}^+$, $CaSO_3^+$, $MgSO_4^+$ | $CaSO_3^+$, $MgSO_4^+$ | - | $C_{10}^+$ | - | gray | |
| 132 | $C_{11}^+$ | - | - | $C_{11}^+$ | - | gray | |
| 133 | $Cs^+$ | $Cs^+$ | - | - | - | orange | |
| 138 | $Ba^+$ | $Ba^+$ | - | - | - | orange | |
| 144 | $C_{12}^+$, $Fe_2O_2^+$ | - | - | $C_{12}^+$ | - | gray | |
| 151 | $C_{12}H_7^+$ | - | - | - | - | | |
| 164 | $K_3PO^+$ | - | - | - | - | | |
| 165 | $Na_3SO_4^+$ | - | $Na_3SO_4^+$ | - | - | blue | |
| 180 | $C_{15}^+$, $K_3PO_2^+$ | - | - | $C_{15}^+$ | - | gray | |
| 181 | $KNa_2SO_4^+$ | - | $KNa_2SO_4^+$ | - | - | | |
| 202 | - | - | - | - | - | | |
| 213 | $K_3SO_4^+$ | $K_3SO_4^+$ | $K_3SO_4^+$ | $K_3SO_4^+$ | $K_3SO_4^+$ | | |
| 217 | - | - | - | - | - | | |
| 222 | - | - | - | - | - | | |
| 223 | - | - | - | - | - | | |
| 224 | $(CaO)_4^+$, $PbO^+$ | - | - | - | - | | |
| 231 | - | - | - | - | - | | |
| 240 | $PbO_2^+$ | - | - | - | - | | |

| (b) anions | | | observed in these particle types | | | marker ions | |
| m/z | potential ions | mineral dust | sea spray | EC | sulfate containing | ALA | LAAP |
|---|---|---|---|---|---|---|---|
| 3 | - | - | - | - | - | | |
| 4 | - | | | | | | |
| 9 | - | - | - | - | - | | |
| 12 | $C_1^-$ | $C_1^-$ | $C_1^-$ | $C_1^-$ | - | gray | |
| 16 | $O^-$ | $O^-$ | $O^-$ | $O^-$ | $O^-$ | | |
| 17 | $OH^-$ | $OH^-$ | $OH^-$ | $OH^-$ | $OH^-$ | | |
| 18 | - | - | - | - | - | | |
| 19 | $F^-$ | $F^-$ | - | - | - | | |
| 23 | $Na^-$ | - | $Na^-$ | - | - | blue | |
| 27 | - | - | - | - | - | | |
| 28 | $CO^-$, $H_2CN^-$ | - | - | - | - | | |
| 29 | - | - | - | - | - | | |
| 32 | $^{32}S^-$, $^{16}O_2^-$ | $^{16}O_2^-$ | $^{16}O_2^-$ | | $^{32}S^-$, $^{16}O_2^-$ | | |
| 34 | $^{34}S^-$, $^{18}O_2^-$ | $^{18}O_2^-$ | $^{18}O_2^-$ | - | $^{34}S^-$, $^{18}O_2^-$ | | |
| 35 | $^{35}Cl^-$ | $^{35}Cl^-$ | $^{35}Cl^-$ | - | - | blue | blue |
| 36 | $C_3^-$ | $C_3^-$ | $C_3^-$ | $C_3^-$ | - | | gray |
| 37 | $^{37}Cl^-$ | $^{37}Cl^-$ | $^{37}Cl^-$ | - | - | blue | blue |
| 40 | - | - | - | - | - | | |
| 42 | $CNO^-$ | $CNO^-$ | $CNO^-$ | $CNO^-$ | $CNO^-$ | | |
| 43 | $AlO^-$, $HCNO^-$ | $AlO^-$, $HCNO^-$ | $HCNO^-$ | - | $HCNO^-$ | orange | |
| 50 | $C_3N^-$ | - | - | - | - | | |
| 60 | $SiO_2^-$, $C_5^-$ | $SiO_2^-$ | - | $C_5^-$ | - | orange | orange |
| 72 | $C_6^-$, $FeO^-$, $(CaO)O^-$ | $FeO^-$, $(CaO)O^-$ | - | $C_6^-$ | - | | gray |
| 76 | $SiO_3^-$, $AlO_2(OH)^-$ | $SiO_3^-$, $AlO_2(OH)^-$ | - | - | - | orange | orange |
| 77 | $HSiO_3^-$ | $HSiO_3^-$ | - | - | - | orange | |
| 78 | - | - | - | - | - | | |
| 96 | $SO_4^-$, $C_8^-$ | $SO_4^-$ | $SO_4^-$ | $SO_4^-$, $C_8^-$ | $SO_4^-$ | green | |
| 97 | $H^{32}SO_4^-$, $C_8H^-$, $H_2PO_4^-$ | $H^{32}SO_4^-$ | $H^{32}SO_4^-$ | $H^{32}SO_4^-$ | $H^{32}SO_4^-$ | green | |
| 99 | $H^{34}SO_4^-$, $SiO_3Na^-$ | $H^{34}SO_4^-$, $SiO_3Na^-$ | $H^{34}SO_4^-$ | $H^{34}SO_4^-$ | $H^{34}SO_4^-$ | green | |
| 101 | - | - | - | - | - | | |
| 106 | - | - | - | - | - | | |
| 115 | $HSO_4(H_2O)^-$, $Na(NO_2)_2^-$ | - | $Na(NO_2)_2^-$ | - | $HSO_4(H_2O)^-$ | | |
| 120 | $C_{10}^-$, $(SiO_2)_2^-$, $NaHSO_4^-$, $(NaCl)NO_3^-$ | - | $NaHSO_4^-$, $(NaCl)NO_3^-$ | $C_{10}^-$ | - | | blue |
| 141 | $CHO_2SO_4^-$, $C_2H_5OSO_4^-$ | - | - | - | $CHO_2SO_4^-$, $C_2H_5OSO_4^-$ | green | |
| 155 | $C_2H_3O_2SO_4^-$, $C_3H_7OSO_4^-$ | - | - | - | $C_2H_3O_2SO_4^-$, $C_3H_7OSO_4^-$ | green | |
| 177 | $HSO_4SO_3^-$, $CH_3(HSO_3)_2^-$ | $HSO_4SO_3^-$, $CH_3(HSO_3)_2^-$ | $HSO_4SO_3^-$, $CH_3(HSO_3)_2^-$ | - | $HSO_4SO_3^-$, $CH_3(HSO_3)_2^-$ | green | |
| 195 | $H(HSO_4)_2^-$, $NH_4HSO_4SO_3^-$ | $H(HSO_4)_2^-$, $NH_4HSO_4SO_3^-$ | - | - | $H(HSO_4)_2^-$, $NH_4HSO_4SO_3^-$ | green | |
| 217 | $MgH(SO_4)_2^-$, $Na(HSO_4)_2^-$, $NH4NaSO_4SO_3^-$ | - | - | - | $MgH(SO_4)_2^-$, $Na(HSO_4)_2^-$, $NH4NaSO_4SO_3^-$ | green | |
| 233 | $KSO_4H_2SO_4-$ | - | - | - | $KSO_4H_2SO_4^-$ | green | |

Accordingly, we update the discussion about our reasoning to choose single ions for correlation coefficient analysis in the method section, lines 282 - 284:

*The advantage of the ion correlation method is that it looks at the correlation of chemical substances rather than whole particle types, which means that fewer initial assumptions have to be made and a cross-particle type approach can be taken.*

Furthermore, we update the explanation of our approach in the results, lines 412 - 427:

*In the following, we present the results of those correlations for the possible particle types inferred from assigning ions to the observed m/z values. A selection of possible ions for each meaningful correlator m/z value  listed in Tab. 1 and the assignment to possible particle types can be found in Tab. S1. The interpretation of particle components and particles types was achieved by the comparison with existing reference mass spectra from both mass spectrometers (see Figs. S9 and S10, panel a), as well as a m/z-to-m/z correlation analysis. The latter method provides information about which ions show a similar time series and thus can either represent isotopes of one element or different molecular fragments of the same original substance. Finally, single meaningful correlators were only assigned to a particle type if other meaningful correlators also indicated the same particle type and if this could be confirmed by both single particle mass spectrometers. When assigning ions to m/z values, it must be taken into account that different ions can be assigned to an integer m/z value, which in turn means that a single m/z value can be assigned to several particle types. This may result, for example, in two different ions*

*of the same m/z value having increased correlations with the INP variables and thus appearing for the same polarity and m/z value for different particle types. Moreover, we also investigate ions with negative correlation coefficients. Furthermore, it should be noted that several particle types can be mixed internally due to long range transport. Therefore, it would not be surprising to find the marker ions in almost all the particle types.. At the end of this section, we also discuss the differences between the two single particle mass spectrometers, and the correlation methods.*

We adjusted the descriptions in section 3.3.1 Sea spray (lines 429 - 433):

*In the analysis of both instruments, we find that sea spray related ions have elevated correlation coefficients with both $[INP]_{-31}$ and $n_s$ (Tab. 1, Tab. S1). The chlorine anions $^{35}Cl^-$ and $^{37}Cl^-$ as well as cations with m/z 46 ($Na_2^+$), 81 ($Na_2^{35}Cl^+$), 83 ($Na_2^{37}Cl^+$) in the LAAPTOF (Tab. 1, panel a) and $^{35}Cl^-$, $^{37}Cl^-$, 23 ($Na^+$), 62 ($Na_2O^+$), 63 ($Na_2OH^+$), 78 ($K_2^+$), 12 ($Mg^{2+}$), 108 ($Na_2NO_3^+$), 181 ($KNa_2SO_4^+$) and 165 ($Na_3SO_4^+$), in the ALABAMA show positive correlation coefficients (r) between 0.31 and 0.57 (Tab. 1, panel b).*

In section 3.3.2 Mineral dust (lines 438 - 443):

*Mineral dust (Tab. 1, Tab. S1) is certainly a particle type that is expected to act as an INP. In general, we find correlation coefficients between ions indicative of mineral dust with both $[INP]_{-31}$ and $n_s$ in the range of 0.32 to 0.60.. For example, the cations m/z 44 ($SiO^+$), 56 ($CaO^+$), 57 ($CaOH^+$), 75 ($CaCl^+$) as well as the anions m/z 60 ($SiO_2^-$), 76 ($SiO_3^-$), and 77 ($HSiO_3^-$) appear in the LAAPTOF correlation table with correlation coefficients between 0.3 and 0.6, in the ALABAMA data set we find cations such as m/z 7 ($Li^+$), 12 ($Mg^{2+}$), 23 ($Na^+$), 24 ($Mg^+$), 39 ($K^+$), 41 ($K^+$), 40 ($Ca^+$), 133 ($Cs^+$), 138 ($Ba^+$) and anions with m/z 43 ($AlO^+$), 76 and 77 ($SiO_3^-$ and $HSiO_3^-$) on the list with correlations coefficients above the threshold.*

In particular, we address now the issue raised by the reviewer regarding elemental carbon in section 3.3.3 (lines 450 - 452):

*However, it should be mentioned that $C_1^-$ (m/z 12) in particular is not a unique feature for elemental carbon, but is also frequently observed in mass spectra of other particle types.*

And update the description about the creation of time series of particle types in lines 523 - 532:

*Similarly, we created a time series for sea spray also from LAAPTOF data, for mineral dust and for elemental carbon (ALABAMA and LAAPTOF), and for sulfate-containing particles (ALABAMA only) using the ions listed in Tab. 1, panel a and b and the color coded marker ions in Tab. S1 (bold marked). The signals at m/z +12, +24, +39, +41 were not considered for the sea spray and mineral dust time series, because these are very common signals across all particle types. The ions at m/z +12 and +24 can be attributed to both carbon and magnesium. Although m/z +12 clearly shows an increased intensity in the EC type compared to the other particle types listed, it is less clear for m/z +24. Therefore, m/z +24 was not used as a marker for any of the particle types. The signals at m/z +39 and +41 are mainly indicative of potassium, which is a common component of mass spectra due to its ionization energy. For example, potassium is occurring in biomass burning particles (e.g., Silva et al., 1999) but is also present in sea water and in mineral dust. Thus, the ions m/z +39 and +41 were not considered here in the analysis.*

Our results are now summarized in the conclusions accordingly in lines 739 - 743:

*Such correlation analysis allows to include also small ion signals which still might represent chemical substances rather than whole particle types, such that fewer assumptions have to be made initially, allowing a cross-particle type approach. Based on our analyses, sodium-, calcium-, silicon- and chlorine-containing ions in particular showed increased correlation with $[INP]_{-31}$ and $n_s$. We concluded that these ions originate from substances that are essentially due to mineral dust and sea salt particles.*

References

Silva, P. J., Liu, D.-Y., Noble, C. A., and Prather, K. A.: Size and Chemical Characterization of Individual Particles Resulting from Biomass Burning of Local Southern California Species, Environmental Science & Technology, 33, 3068-3076, 10.1021/es980544p, 1999.